# Beyond the 'big four': Venom profiling of the medically important yet neglected Indian snakes reveals disturbing antivenom deficiencies

**R. R. Senji Laxme[1]☯, Suyog Khochare[1]☯, Hugo Francisco de Souza[1]☯, Bharat Ahuja[1], Vivek Suranse[1], Gerard Martin[2], Romulus Whitaker[3], Kartik Sunagar[1]***

**1** Evolutionary Venomics Lab. Centre for Ecological Sciences, Indian Institute of Science, Bangalore, Karnataka, India, **2** The Gerry Martin Project. Survey #1418/1419 Rathnapuri, Hunsur, Karnataka, India, **3** Madras Crocodile Bank Trust/Centre for Herpetology, Mamallapuram, Tamil Nadu, India

☯ These authors contributed equally to this work.
* ksunagar@iisc.ac.in

**Data Availability Statement:** All relevant data are within the manuscript and its Supporting Information files.

## Abstract

### Background

Snakebite in India causes the highest annual rates of death (46,000) and disability (140,000) than any other country. Antivenom is the mainstay treatment of snakebite, whose manufacturing protocols, in essence, have remained unchanged for over a century. In India, a polyvalent antivenom is produced for the treatment of envenomations from the so called 'big four' snakes: the spectacled cobra (*Naja naja*), common krait (*Bungarus caeruleus*), Russell's viper (*Daboia russelii*), and saw-scaled viper (*Echis carinatus*). In addition to the 'big four', India is abode to many other species of venomous snakes that have the potential to inflict severe clinical or, even, lethal envenomations in their human bite victims. Unfortunately, specific antivenoms are not produced against these species and, instead, the 'big four' antivenom is routinely used for the treatment.

### Methods

We characterized the venom compositions, biochemical and pharmacological activities and toxicity profiles (mouse model) of the major neglected yet medically important Indian snakes (*E. c. sochureki*, *B. sindanus*, *B. fasciatus*, and two populations of *N. kaouthia*) and their closest 'big four' congeners. By performing WHO recommended *in vitro* and *in vivo* preclinical assays, we evaluated the efficiencies of the commercially marketed Indian antivenoms in recognizing venoms and neutralizing envenomations by these neglected species.

### Findings

As a consequence of dissimilar ecologies and diet, the medically important snakes investigated exhibited dramatic inter- and intraspecific differences in their venom profiles. Currently marketed antivenoms were found to exhibit poor dose efficacy and venom recognition

**Funding:** KS was supported by the following grants: the Department of Biotechnology-IISc Partnership Program (http://dbtindia.gov.in/), DST-INSPIRE Faculty Award (DST/INSPIRE/04/2017/000071, http://online-inspire.gov.in/), and the DST-FIST (SR/FST/LS-II/2018/233, http://www.fist-dst.org/). This work was also funded with UK aid from the UK Department for International Development. The views expressed do not necessarily reflect the UK Government's official policies. RW was supported by USV Private Limited (http://www.usvindia.com). HFdS was supported by the DST INSPIRE Fellowship (IF170819, http://online-inspire.gov.in/). The funders had no role in study design, data collection and analysis, decision to publish, or preparation of the manuscript.

**Competing interests:** The authors have declared that no competing interests exist.

potential against the 'neglected many'. Premium Serums antivenom failed to neutralise bites from many of the neglected species and one of the 'big four' snakes (North Indian population of *B. caeruleus*).

## Conclusions

This study unravels disturbing deficiencies in dose efficacy and neutralisation capabilities of the currently marketed Indian antivenoms, and emphasises the pressing need to develop region-specific snakebite therapy for the 'neglected many'.

## Author summary

Snakebite is a 'neglected tropical disease' that majorly affects the rural populations in developing countries. India bears the brunt of snakebites with over 46,000 deaths and 140,000 disabilities, annually. A significant number of these bites are attributed to the widely distributed 'big four' snakes, namely spectacled cobra (*Naja naja*), common krait (*Bungarus caeruleus*), Russell's viper (*Daboia russelii*), and saw-scaled viper (*Echis carinatus*). The commercial antivenoms marketed in India are only manufactured against these four species, while neglecting many other medically relevant snakes with restricted geographic distribution. Snakebite pathology is dependent on the venom composition of the population/species, which can, in turn, vary intra- and inter-specifically. Though this variation severely limits the cross-population/species antivenom efficacy, envenomations by the neglected snakes in India are treated with the 'big four' antivenom. Therefore, to unravel the underlying venom variability, we investigated venom proteomic, biochemical/pharmacological and toxicity profiles of the major neglected Indian snakes and their 'big four' relatives. To assess the effectiveness of the 'big four' antivenom in treating bites from these neglected snakes, we performed preclinical experiments, which revealed alarming inadequacies of the commercial antivenoms. Our findings accentuate the compelling necessity for the innovation of highly efficacious next-generation snakebite therapy in India.

## Introduction

Snakebite is amongst the foremost neglected tropical diseases (NTD) plaguing the world today. The World Health Organization (WHO) estimates over 2.7 million annual cases of clinical illness following snake envenomation, with over 500,000 of those resulting in death or permanent disabilities [1]. The impact of snakebite, however, is grossly underestimated. Since the majority of bite victims in Southeast Asia and Africa belong to rural subsistence farming families that fall below the poverty line [2], and the incidence of snakebites is higher in young males [3], snakebite results in severe socioeconomic repercussions in developing nations. Moreover, the majority of envenomated individuals are the primary bread winners of their families and, thus, snakebites potentially cripple entire families.

India is the global hotspot for snakebites, where 46,000 deaths and 140,000 disabilities result from snakebites annually [1]. In India, a polyvalent antivenom, the only scientifically proven antidote to the toxic effects of snakebite, is produced against the 'big four' snake species, namely, the spectacled cobra (*Naja naja*), the common krait (*Bungarus caeruleus*), Russell's viper (*Daboia russelii*), and the saw-scaled viper (*Echis carinatus*). Given their fairly large distribution across the Indian subcontinent, these snakes are responsible for a majority of

medically important snakebites in the country [1]. India is home to many other snake species, including various species of cobras, kraits, saw-scaled vipers, sea snakes, and pit vipers, which are capable of delivering clinically severe envenomations [4]. Unfortunately, however, a specific antivenom against these snakes is not produced. Moreover, the majority of economically weaker snakebite victims resort to the easily affordable traditional faith healers and charlatans, instead of hospitals. This, together with the inability to identify snake species responsible for bites, has resulted in envenomation figures from both the 'big four' and 'non-big four' species being largely underestimated and undocumented in many states, such as in Northeast India. Consequently, no attempt to develop specific antivenom for these 'neglected many' has been hitherto undertaken. Sadly, however, envenomation by these neglected snakes have been, and are continued to be treated using the conventional polyvalent antivenom. Since snakes can exhibit significant inter- and intraspecific venom variation, polyvalent antivenom raised against the 'big four' species, may be incapable of fully neutralising the clinical symptoms of the other, neglected snake envenomations.

In this study, we unravel the venom protein composition, biochemical and pharmacological activities, and the toxicity profiles of four neglected snake species–two populations of the monocled cobra (*Naja kaouthia*), Sochurek's saw-scaled viper (*Echis carinatus sochureki*), the banded krait (*Bungarus fasciatus*), and the Sind krait (*Bungarus sindanus*), as well as their closest 'big four' counterparts, namely *N. naja*, *E. carinatus*, and *B. caeruleus*, respectively. Using *in vitro* and *in vivo* experiments, we further evaluate the binding and neutralising efficiencies of four commercial polyvalent Indian antivenoms in treating bites from both the neglected species and their respective 'big four' relatives. Our results highlight stark inter- and intraspecific differences in venom composition, biochemical and pharmacological effects, and toxicity profiles, and the poor cross neutralising capabilities of commercial Indian antivenoms, thereby, emphasising the urgent and pressing need to look 'beyond the 'big four' snakes' and develop specific snakebite therapy against the 'neglected many'.

## Methods

### Sampling permits and snake venoms

Venom samples, pooled from 2–17 snakes, were collected from the following States with appropriate permissions from the respective State Forest Departments (Fig 1A and 1B): Maharashtra [Desk-22(8)/WL/Research/CR-60(17–18)/2708/2018-19)], Arunachal Pradesh [CWL/Gen/13(95)/2011-12/Pt.VI/890-94)], Rajasthan [P.3(3)Forest/2004], Punjab [#3615;11/10/12], and West Bengal [386/WL/4R-6/2017]. Snake species were identified by expert herpetologists, Romulus Whitaker and Gerard Martin, and venom samples were collected by them, using WHO protocols, snap frozen, lyophilized, reconstituted in ultra-pure water, and were stored at -80$^\circ$ C until further use. Details of individual venom samples are provided in the S1A Table.

### Ethical statement

Venom Median Lethal Dose (LD$_{50}$) and Median Effective Dose (ED$_{50}$) assays were performed at the Central Animal Facility, Indian Institute of Science, Bangalore (Registration number 48/GO/ReBi/SL/1999/CPCSEA; 11-03-1999) using WHO-approved preclinical murine assays. Experiments were performed on male CD1 mice (18–22 g) with due approval from the Institutional Animal Ethics Committee (IAEC) Indian Institute of Sciences (IISc), Bangalore (CAF/Ethics/642/2018; CAF/Ethics/643/2018; CAF/Ethics/687/2019). Our protocols for animal care and handling were in accordance with the guidelines of the Committee for the Purpose of Control And Supervision of Experiments on Animals (CPCSEA). To greatly reduce the number of mice consumed in experiments, only a single antivenom was subjected to *in vivo* testing.

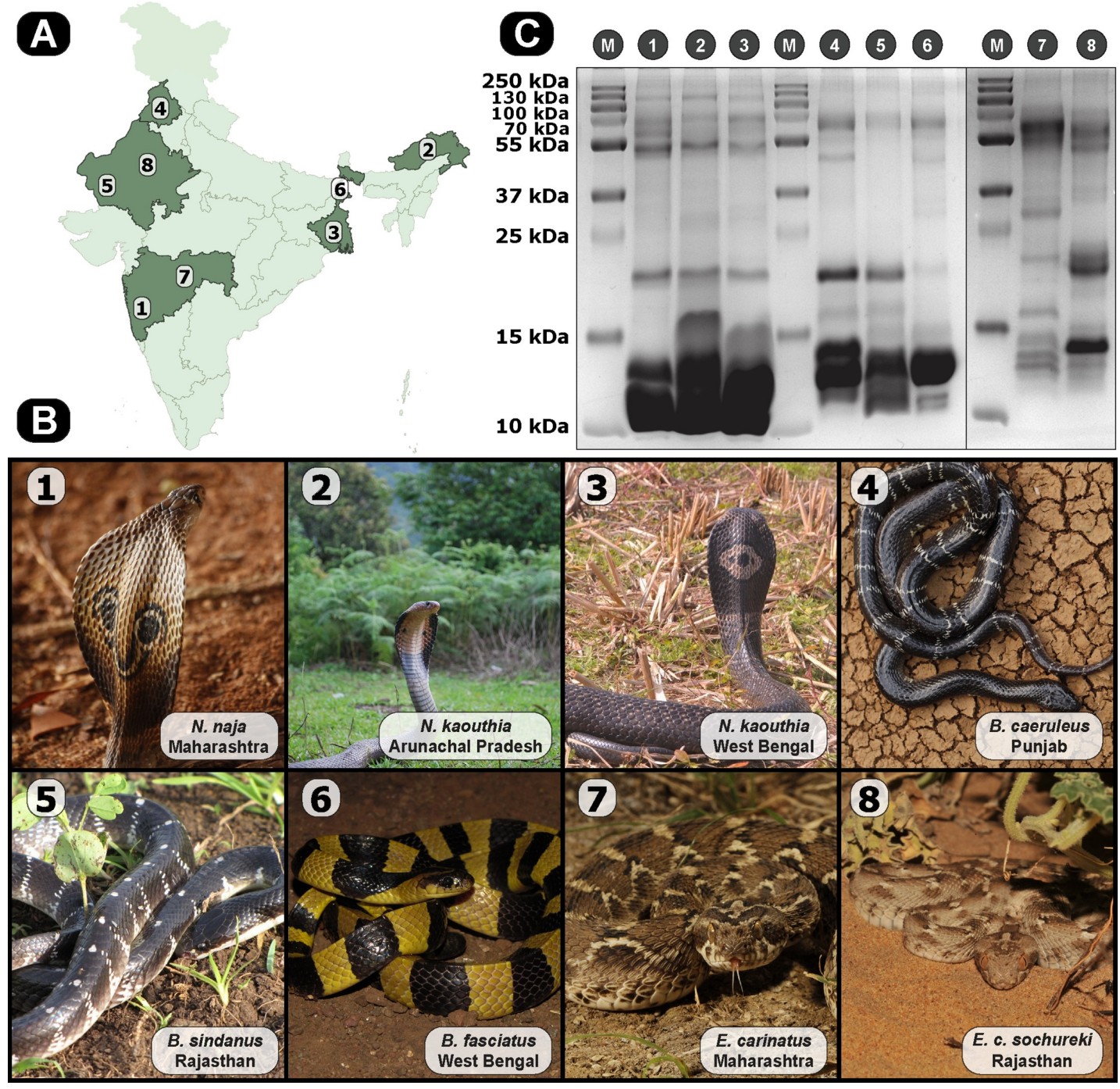

**Fig 1. Medically important Indian snakes and their venom profiles.** (A) and (B) are indicative of sampling locations and photographs of neglected snakes and their 'big four' counterparts, respectively. The map of India shown in panel (A) was prepared using QGIS 3.8 [20]. (C) venom samples, normalized for protein content (12–15 μg), were loaded onto 12.5% SDS-PAGE. Lane numbers indicate corresponding species (M: pre-stained protein ladder).

Mice were humanely euthanized in a $CO_2$ chamber by trained professionals after the completion of experiments. Ethical permits were also obtained from the Institute Human Ethical Committee (IHEC; Permit No: 5–24072019) for testing the coagulopathic effects of venom on the blood collected from healthy volunteers with written consent.

## Antivenoms

Marketed polyvalent antivenoms, manufactured against the 'big four' species, were acquired from Bharat Serums and Vaccines Ltd., Haffkine Biopharmaceuticals Corporation Ltd., Premium Serums & Vaccines Pvt. Ltd. and VINS Bioproducts Ltd., in lyophilized form (S1B Table). For the estimation of total protein content, one vial of the respective antivenom was reconstituted as per the manufacturer's recommendations.

## Protein quantification

The protein contents of crude venom samples were determined using a modified Bradford method [5]. Lyophilized venom (3 μL), reconstituted in ultrapure water, was incubated with 250 μL of the Bradford Reagent (Sigma-Aldrich, USA) in triplicates at room temperature (RT) for 20 minutes, followed by the estimation of protein concentrations in an Epoch 2 microplate spectrophotometer (BioTek Instruments, Inc., USA) at 595 nm wavelength. Pure Bovine Serum Albumin (BSA; Sigma-Aldrich, USA) and purified naïve horse immunoglobulin G (IgG; Bio-Rad Laboratories, USA) of known concentration were used as standards for the estimation of protein content in venom and antivenom samples, respectively.

## Proteomics

**SDS-PAGE.** Reconstituted venom samples were subjected to 15% sodium dodecyl sulphate–polyacrylamide gel electrophoresis (SDS-PAGE) under reducing conditions [6], followed by the staining of the gels with Coomassie Brilliant Blue R-250 (Sisco Research Laboratories Pvt. Ltd, India), and visualization in an iBright CL1000 (Thermo Fisher Scientific, USA).

**Tandem mass spectrometry.** For the characterization of proteomic profiles, venom samples were subjected to electrospray ionization liquid chromatography tandem mass spectrometry (ESI-LC-MS/MS) analysis. The venom samples were reduced using Dithiothreitol (DTT, Sigma-Aldrich, USA) and alkylated using iodoacetamide (Sigma-Aldrich, USA), followed by digestion with MS grade Trypsin enzyme (2 μg/μl; Promega Corporation, USA). These samples were then subjected to tandem mass spectrometry analysis in an Ultra-Performance Liquid Chromatography (1290 Infinity II, Agilent Technologies, USA), coupled with an ESI-QTOF mass spectrometer (AdvanceBio 6545XT, Agilent Technologies, USA). For the identification of toxin classes present in the venom, PEAKS Studio X (Bioinformatics Solutions Inc.) was used to independently search the raw MS/MS spectra against NCBI's 'non-redundant' (www.ncbi.nlm.nih.gov) and Uniprot's SwissProt databases (December 2018; www.uniprot.org). The following set of parameters were used for analyses: precursor and fragment mass error tolerance of 10 ppm and 0.02 Da, respectively; monoisotopic mass search; Carbamidomethylation as fixed modification; Acetylation (N-terminus) and Oxidation (M) as variable modifications. Common contaminants were eliminated by including sequences from the common Repository of Adventitious Proteins (www.thegpm.org/crap) into the search database. To estimate relative abundance, the area under the curve of each peptide feature at the same m/z and retention time as the MS/MS scan, was calculated in PEAKS Studio X.

## Biochemical assays

**Phospholipase $A_2$ ($PLA_2$) assay.** $PLA_2$ assays were performed using a previously described turbidometric assay that was slightly modified to work in microtiter plates [7]. Egg-yolk substrate solution was prepared by carefully separating yolk from a fresh chicken egg, dissolving it in 0.9% NaCl solution, and adjusting absorbance at 740 nm to 1 optical density (OD)

using 20mM Tris-Cl buffer (pH 7.4). Time-dependent kinetic assays were performed in triplicate using various concentrations of crude venom samples (0.01 μg, 0.1 μg, 0.5 μg, and 1 μg) in 20mM Tris-Cl buffer (pH 7.4). 250 μL of the egg yolk solution was added to the samples, and absorbance read at 740 nm over a 60 min period in an Epoch 2 microplate spectrophotometer (BioTek Instruments, Inc., USA). The unit activity of PLA$_2$ activity is described as the amount of crude venom PLA$_2$ which decreases the turbidity of the solution by 0.01 absorbance unit per minute at 740 nm [8].

**L-amino acid oxidase assay.** L-amino acid oxidase (LAAO) assay, with L-leucine as substrate, was carried out in triplicate in microtiter plates with a slight modification to the previously described method [9]. The reaction mixture comprised of 10 μg crude venom and 90 μL of the substrate solution, which, in turn, contained 50mM Tris-HCl buffer (50mM), L-leucine (5mM), horseradish peroxidase (5 IU/mL), and o-phenylenediamine dihydrochloride (2mM). The reaction mixture was incubated at 37˚ C for 60 minutes, and the reaction was stopped by the addition of 2M H$_2$SO$_4$. Absorbance at 492 nm was measured using an Epoch 2 microplate spectrophotometer.

**Hyaluronidase activity.** Hyaluronidase activity was measured using a turbidometric method described previously, with modifications for 96-well microtiter plates [10]. The assay mixture contained acetate buffer (0.2M sodium acetate-acetic acid, 0.15M NaCl, pH 6.0), 1 mg/mL hyaluronic acid (Sigma-Aldrich, USA), and 2.5 μg crude venom samples in a final volume of 100 μL. A standard curve with different concentrations of hyaluronic acid was used to determine the optimum concentration of the enzyme, and acetate buffer was used as the blank. Purified hyaluronidase from sheep testicles (Sigma-Aldrich, USA) was used as the positive control. The reaction mixture was incubated at 37˚ C for 30 minutes, and the reaction was stopped by adding 200 μL of stop solution, containing 2.5% (w/v) cetyltrimethylammonium bromide (CTAB) dissolved in 2% (w/v) NaOH. The absorbance was read at 400 nm using an Epoch 2 microplate spectrophotometer. One turbidity reduction unit (TRU) was defined as the amount of enzyme required to reduce 50% of the turbidity in the reaction. The results were expressed as TRU/mg$^{-1}$min$^{-1}$ in relation to the positive control.

**Snake venom protease assay.** Snake venom protease activity of the crude venom was assayed using a protocol described previously, with modifications [11]. Azocaesin (SRL, India) in 0.05 M Tris-HCl (pH 8) served as the substrate for the reaction. The substrate and crude venom (8:1) were incubated at 37˚ C for 90 minutes. The reaction was stopped by adding 5% (v/v) trichloroacetic acid and centrifuged at 1000 x g for 5 minutes. The supernatant was carefully collected and equal volumes of 0.5 M NaOH was added, and the absorbance was measured at 440 nm using Epoch 2 microplate spectrophotometer. The relative activity of crude venoms to the purified protease from bovine pancreas (Sigma-Aldrich, USA; positive control) was plotted. All reactions were performed in triplicates.

**Fibrinogenolytic assay.** The fibrinogenolytic activity of the venoms were visualised using fibrinogen from human plasma (Sigma-Aldrich, USA) and applying a modified version of a method previously described [12]. The reaction mixture contained 15 μg of fibrinogen, dissolved in phosphate buffered saline (pH 7.4, PBS), along with known concentrations of venoms (1.5 μg) and was incubated at 37˚ C for 60 minutes. At the end of the incubation time, equal volumes of loading dye (1M Tris-HCl pH 6.8, 50% Glycerol, 0.5% Bromophenol blue, 10% SDS, and 20% β-mercaptoethanol) was mixed with the aforementioned reaction mixture and heated at 70˚ C for 10 minutes to arrest the reaction. This was followed by electrophoresis on a 15% SDS-PAGE. After staining with Coomassie Brilliant Blue R-250, the gels were visualized using an iBright CL1000.

**DNase assay.** To determine the DNase activity of venom, 500 ng of calf thymus DNA (SRL, India) was incubated with a known concentration of crude venom and PBS at 37˚ C for

60 minutes. The assay mixture was subjected to 0.8% agarose horizontal gel electrophoresis and imaged using an iBright CL1000. DNase I from bovine pancreas (15 U; Sisco Research Laboratories Pvt. Ltd, India) was used as the positive control. The protocol was standardized by modifying a method described earlier [13].

**ATPase and ADPase assays.** ATPase and ADPase assays, were carried out in triplicates with adenosine triphosphate (ATP) and adenosine diphosphate (ADP) substrates, respectively, following a modified method described earlier [14]. In brief, reaction mixtures containing venom (34 µg) and incubation buffer (1 mg/mL ATP or ADP, 50 mM Tris-Cl pH 7.4, 3.8 mM $MgCl_2$) in a total volume of 200 µl, were warmed at $37^0$ C for 1 and 3 hours. Following incubation, a coloring reagent (10% ascorbic acid and 0.42% ammonium molybdate in 1N $H_2SO_4$) was added, and the mixture was microwaved for 30 seconds. 0.3M trichoroacetic acid was added to stop the reaction, and the absorbance was measured at 820 nm using an Epoch 2 microplate spectrophotometer.

**Blood coagulation assays.** To understand the effect of venom on the blood coagulation pathway, prothrombin time (PT) and activated partial thromboplastin time (aPTT) were assayed, which correspond to the time taken for the appearance of the first fibrin clot via extrinsic and intrinsic coagulation pathways, respectively. Blood donated by healthy male volunteers, collected in a tube with 3.2% sodium citrate, was centrifuged at 2400 x g for 5 mins at 4˚ C. The supernatant or the Platelet Poor Plasma (PPP) was collected and mixed with different concentrations of venom. These pharmacological assays were conducted using diagnostic kits, following the manufacturer's protocols (Agappe diagnostic Ltd, Ernakulam). The international normalized ratio (INR) for PT was calculated using the following formula, where International Sensitivity Index (ISI) is provided by the manufacturer.

$$INR = \left( \frac{PT_{Test}}{PT_{Control}} \right)^{ISI}$$

**Haemolytic assay.** Haemolytic activities of venoms were assayed in triplicates using the *in vitro* method described earlier with slight modifications [15]. Following the separation of PPP from the whole blood, the red blood cells (RBC) were washed with 1X PBS solution (pH 7.4) and centrifuged at 15,000 x g for 10 mins at 4˚ C. The RBC pellet was resuspended in PBS and incubated with different concentrations of crude venoms (5, 10, 20, and 40 µg) at $37^0$ C for 24 hours. These reaction mixtures were then centrifuged at 15,000 x g for 5 mins at $4^0$ C, followed by the measurement of absorbance of the supernatant at 595 nm using an Epoch 2 microplate spectrophotometer. The RBC suspension with 1% Triton X and PBS alone were treated as positive and negative controls, respectively.

## Endpoint ELISA

To evaluate capabilities of the currently marketed Indian polyvalent antivenoms in recognising and binding to epitopes on venom proteins, we conducted endpoint titration enzyme-linked immunosorbent assay (ELISA), following a protocol described earlier with slight modifications [16]. Lyophilized venoms were reconstituted in ultrapure molecular grade water to a final concentration of 1 mg/mL. Venom samples (100 ng) in carbonate buffer (pH 9.6) were coated on the 96-well plates and incubated overnight at 4˚ C. The wells were rinsed 6 times with Tris-buffered saline (0.01 M Tris pH 8.5, 0.15 M NaCl), 1% Tween 20 (TBST), and incubated for 3 hours at RT with 5% skimmed milk in TBST as the blocking solution. Plates were washed with TBST and incubated overnight with antivenoms at 4˚ C. All four antivenoms (1 mg/mL) were added at an initial dilution of 1:4, followed by increments of 1:5 serial dilutions. After washing,

horseradish peroxidase (HRP)-conjugated, rabbit anti-horse secondary antibody (Sigma-Aldrich, USA), diluted at a ratio of 1:1000 in PBS was added to all wells, and the plate was incubated at RT for 2 hours. After washing, 100 μL of 2,2/-azino-bis (2-ethylbenzthiazoline-6-sulphonic acid) substrate solution (Sigma-Aldrich, USA) was added and incubated at RT for 40 minutes. The optical density was measured at a wavelength of 405 nm. The assay was performed in triplicates and the mean values were used for calculations. Purified equine IgG (Bio-Rad Laboratories, USA) was used as the negative control, and absorbance values above the cut off, calculated as the mean absorbance of the negative control plus two times the standard deviation, were considered for comparisons.

### Western blotting

To visualise venom-antivenom (antigen-antibody) interactions, we performed immunoblotting experiments. Following the electrophoretic separation of venom proteins (25 μg) on a 4–20% gradient SDS-PAGE gel, protein bands were electroblotted onto a 0.45 μm nitrocellulose membrane following the manufacturer's protocol (Bio-Rad Laboratories, USA). The membrane was reversibly stained with Ponceau S (Sigma-Aldrich, USA) to confirm protein transfer, and then incubated overnight with blocking solution (5% non-fat milk in TBST). The membrane was then incubated overnight with the marketed antivenom, diluted to 1:200 concentrations in the blocking solution, following which it was incubated with HRP-conjugated, rabbit anti-horse secondary antibody (1:2000 dilution) at RT for 2 hours. Following the addition of substrate solution containing 0.05% DAB (3,3-diaminobenzidine; Sigma-Aldrich, USA) reagent in PBS and 0.024% hydrogen peroxide ($H_2O_2$; Thermo Fisher Scientific, USA), the membrane was imaged. All of these aforementioned steps included intermittent washing steps (n = 6) with TBST [16].

### *In vivo* venom lethality (Median Lethal Dose—$LD_{50}$)

Toxicity profiles of the venoms were determined using male CD-1 mice (18–22 g) as per previously described methods and WHO guidelines [17]. Briefly, groups of five male CD-1 mice were intravenously injected into the tail vein with five different concentrations of venom diluted in 500 μL of 0.9% (w/v) physiological saline. After 24 hours of observation, death and survival patterns were recorded, and the $LD_{50}$ value was calculated using Probit statistics [18].

### *In vivo* venom neutralization (Median Effective Dose—$ED_{50}$)

The preclinical efficacy of the Premium Serums antivenom in neutralising venoms of the neglected species was determined by conducting $ED_{50}$ experiments, where $ED_{50}$ is defined as the amount of antivenom that safeguards 50% of the test population [17]. Four different antivenom dilutions were challenged against venom concentration five times that of the $LD_{50}$ value determined above for each venom. Venoms were incubated with four dilutions of antivenom at $37^0$ C for 30 minutes, followed by intravenous injection into tail vein of five CD-1 mice per dilution. An additional group of mice (n = 5), which was injected with the respective $LD_{50}$ of venom, served as the positive control. The median effective dose of antivenom was calculated using Probit analysis, after 24 hours of observation. For *N. kaouthia* population from Arunachal Pradesh, the two highest antivenom concentrations were completely ineffective against the 5x $LD_{50}$ venom dose and, therefore, we repeated the experiment using a reduced venom challenge dose equivalent to 3x $LD_{50}$. Following the completion of these experiments, antivenom potency was expressed in milligrams of venom neutralised per millilitre of

antivenom using the following equation.

$$\text{Antivenom potency} = \frac{(\text{CD} - 1)}{\text{ED}_{50} \ (\mu l)} X \ \text{LD}_{50} \ \text{of venom} \ (\mu g/\text{mouse})$$

where, CD = number of challenge doses used in the $\text{ED}_{50}$ experiment [19].

## Statistical analyses

For statistical comparisons throughout, one-way and two-way ANOVA were conducted, followed by Dunnett's multiple comparison test in GraphPad Prism 8 (GraphPad Software, La Jolla California USA, www.graphpad.com).

## Results

### Venom profiles

The venom protein composition of venoms of the medically important but neglected Indian snake species and their 'big four' counterparts were elucidated by SDS-PAGE and tandem mass spectrometry. Venoms of these snakes were found to contain a wide range of low- and high-molecular weight toxins, and exhibited profound compositional diversity both inter- and intra-specifically. SDS-PAGE venom profiles of snakes from the genera *Naja*, *Bungarus* and *Echis* revealed several unique bands of diverse molecular weights and intensities, highlighting the primary differences in their venom composition and relative abundance, respectively (Fig 1C). While subtle differences in band intensities were observed in the venoms of *Naja spp.*, stark compositional differences were observed for certain toxins in *Bungarus spp.* and *Echis* subspecies. In order to get a better understanding of the observed proteomic diversity, the venoms were subjected to tandem mass spectrometric analyses.

These analyses resulted in the identification of 51–59 proteins in the crude venoms of each *Naja spp.* (Fig 2A; S2A–S2C Table), with the neurotoxic (short- and long-chain α-neurotoxins and muscarinic toxins) three-finger toxin (3FTx), comprising over 40% of the venom proteome, being the most abundant toxin class. Neurotoxic 3FTxs belong to the 3FTx superfamily of non-enzymatic venom proteins that consist of short-length (60–75 residues) toxins, capable of exerting a broad range of clinical effects [21]. Surprisingly, more than three quarters of the venom of *N. kaouthia* from West Bengal contained neurotoxic 3FTx (86%). Contrastingly, the Arunachal Pradesh population of *N. kaouthia* was found to secrete large amounts of cytotoxic 3FTxs that constituted nearly 19% of its venom. Venoms of *N. naja* and *N. kaouthia* (WB), on the other hand, comprised of a very smaller fraction of cytotoxic 3FTxs (~4–5%). Other major venom components detected in *Naja spp.* include the cobra venom factor (CVF), PLA$_2$, and cysteine-rich secretory proteins (CRISP), though, their relative abundances varied substantially between species (Fig 2A; S2A–S2C Table). For example, when compared to the two populations of *N. kaouthia* in Arunachal Pradesh and West Bengal, *N. naja* was found to secrete more than double the amount of CVFs (17% vs 7.9% and 3.2%, respectively), and several fold increased amounts of PLA$_2$ (14.5% vs 0.8% and 3.1%, respectively). Similarly, *N. kaouthia* (AR) venom was found to contain 6–11 times the number of CRISPs (13.1%) when compared to its 'big four' counterpart (2.4%) and the *N. kaouthia* West Bengal population (1.2%). The other notable difference observed in the venom proteomes of these species was related to Kunitz peptides. While nearly 6% of the *N. naja* venom consisted of Kunitz peptides, none were detected in the two *N. kaouthia* populations. Inter- and intraspecific differences in the composition and expression of various 'minor' venom components (<5% of the proteome), including 5'-nucleotidase, acetylcholinesterase, C-type lectins, LAAO, venom nerve growth

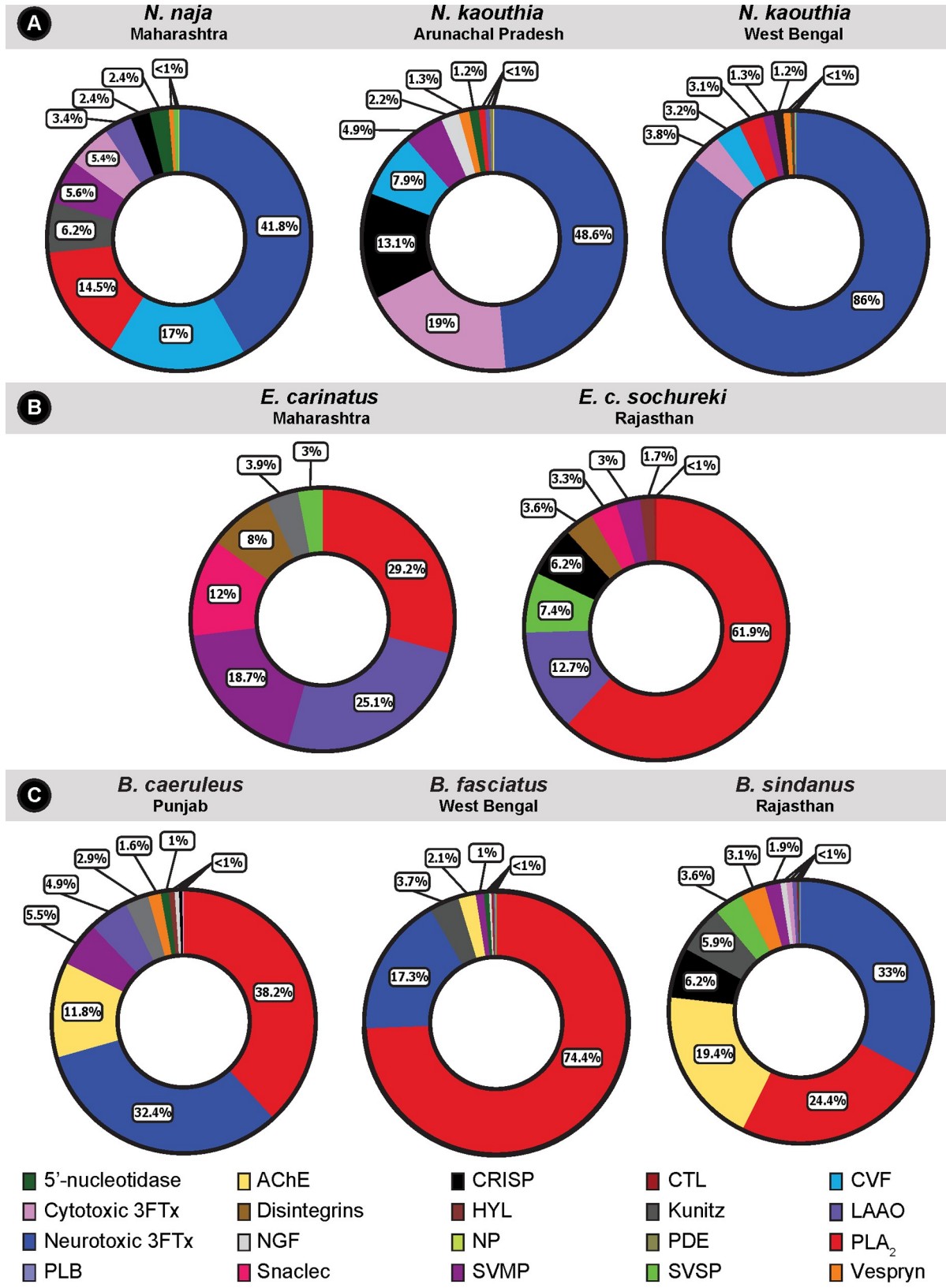

**Fig 2. Comparative venom profiles of the medically important Indian snakes.** Venom compositions of *Naja* (A), *Echis* (B) and *Bungarus* (C) species are depicted as pie charts, and the relative composition of toxins are indicated in percentiles. A unique colour key for individual toxins is provided.

factors, natriuretic peptides, phospholipase B (PLB), snake venom serine proteases (SVSP), and vespryn, were also noted in these snakes (Fig 2A; S2A–S2C Table).

Venoms of *Echis* subspecies predominantly consisted of 7 to 9 toxin families, including $PLA_2$, snake venom metalloproteinase (SVMP), LAAO, SVSP, and snaclec (Fig 2B; S2D and S2E Table). Surprisingly, while venom composition remained fairly similar, notable differences in relative abundance of various toxins were observed between the two subspecies. $PLA_2$ constituted a major proportion (61.9%) of toxins in *E. c. sochureki* venom which was two-fold the amount detected in *E. carinatus* (29.2%). On the contrary, LAAO (25.1%), SVMP (18.7%), snaclec (12%) and disintegrin (8%) were relatively more abundant in *E. carinatus* venom (Fig 2B; S2D and S2E Table). While we detected relatively high amounts of SVSP in *E. c. sochureki* (7.4%) in comparison to *E. carinatus* (3%), CRISPs (6.2%) were only detected in the former subspecies. Hyaluronidase, one of the minor components of snake venoms, commonly known as the 'spreading factor' [22], was also identified in the proteome of both *E. carinatus* (3.9%) and *E. c. sochureki* (1.7%).

The proteomic analyses of crude venoms of the *Bungarus spp.* identified between 37 to 63 toxins from 12 protein families (Fig 2C; S2F–S2H Table). Similar to the cobra species analysed here, the venoms of *Bungarus spp.* were also found to be dominated by neurotoxic 3FTxs and $PLA_2s$. While $PLA_2$ constituted nearly three quarters of *B. fasciatus* venom (74.39%), far lower abundances were observed with the venoms of *B. caeruleus* (38.16%) and *B. sindanus* (24.37%). The relative abundance of neurotoxic 3FTxs was found to be fairly similar across the *Bungarus spp.* and ranged between 17.30–32.97%. Acetylcholinesterases (AChEs) constituted the third major family of venom proteins in *B. caeruleus* and *B. sindanus* (11.76 and 19.42%, respectively), but exhibited considerably lower abundance in *B. fasciatus* (2.14%). Limited amounts of SVMP and LAAO were detected in the venom of *B. caeruleus* (5.54% and 4.86%, respectively), and only trace amounts of these toxins were identified in *B. sindanus* (1.91% and 0.53%, respectively) and *B. fasciatus* (1.01% and 0.13%, respectively). In contrast, higher amounts of CRISPs and Kunitz-type serine protease inhibitors were identified in the venom of *B. sindanus* (6.23% and 5.91%, respectively), relative to *B. caeruleus* (0.43% and 2.87%) and *B. fasciatus* (none detected and 3.68%). Several other classes of toxins, including 5'-nucleotidase, hyaluronidase, phospholipase B, venom nerve growth factor (NGF) and vespryn were also identified in the venom proteome of *Bungarus spp.*

### Venom biochemistry

Compositional differences in snake venoms have been shown to have profound implications on the pathologies that present in snakebite victims [23–25]. Therefore, we evaluated the biochemical/functional activities of the venoms under investigation in a variety of *in vitro* biochemical assays.

**$PLA_2$ assay.** $PLA_2$ is amongst the most important snake venom toxin superfamilies, and the venoms of many Elapidae and Viperidae snakes are enriched with this toxin type [26]. The abundance and type of $PLA_2$ toxins can substantially alter the clinical profile of the venom [27]. Therefore, we conducted enzymatic $PLA_2$ assays on the venoms of the neglected snakes and the related 'big four' species. Our results revealed significant differences in $PLA_2$ catalytic activities (Fig 3A; S1 Fig), where, at the highest venom concentrations tested (0.5 and 1.0 μg), *N. naja* and *N. kaouthia* (AR) showed the highest activities. Interestingly, at very low venom

concentrations (0.01 and 0.1 μg), *N. kaouthia* from West Bengal exhibits slightly greater activity than both *N. naja* and *N. kaouthia* (AR), highlighting its relatively superior catalytic efficiency. At the lowest venom concentration (0.01 μg), PLA$_2$s in *Bungarus spp*. did not show any catalytic activity. However, on increasing the concentration (0.5 μg & 1 μg), *B. sindanus* and *B. caeruleus* exhibited activities that were comparable to those of *Naja spp*. *Echis* venoms exhibited very little to no PLA$_2$ enzymatic activity at lower venom concentrations (0.01 μg and 0.1 μg), and at the highest concentrations (0.5 and 1 μg), *E. carinatus* venom exhibited relatively more activity when compared to *E. c. sochureki*. All these comparisons were statistically significant ($p < 0.05$).

**L-amino acid oxidase (LAAO) assay.** LAAO is amongst the prominent enzymatic components found in venoms of snakes, where they contribute to toxicity by exerting pharmacological effects, such as cytotoxicity, haemorrhage, induction of apoptosis, and/or inhibition of platelet aggregation [28–30]. Considering their role in shaping clinical profiles of envenomation, we assayed LAAO activities of venoms of the neglected species and their closest 'big four' counterparts. Our experiments revealed significant differences ($p < 0.05$) in LAAO enzymatic activities (Fig 3B). While both *N. kaouthia* venom samples exhibited little to complete absence of LAAO activity, *N. naja* venom catalysed the oxidative deamination of significant amounts of L-amino acids. In contrast, all *Bungarus spp*. and *Echis sub spp*. exhibited LAAO activities, albeit in varying degrees. While both *B. caeruleus* and *B. fasciatus* showed higher LAAO activity than *B. sindanus*, *E. c. sochureki* showed significantly higher enzymatic activity in comparison to *E. carinatus*.

**Hyaluronidase assay.** Following envenomation, hyaluronidase activity is considered critical for the diffusion of toxins from the bite site into the circulatory system. Given the importance of hyaluronidase as a spreading factor and its potential to alter clinical profiles of envenomation [31], we assayed the hyaluronidase activities for the venoms of the neglected species and their 'big four' counterparts. Our results revealed significant differences ($p < 0.05$) in hyaluronidase activities (Fig 3C). While both *N. naja* and *N. kaouthia* (AR) showed comparable hyaluronidase activities, *N. kaouthia* (WB) showed the highest activity among the *Naja* venom samples tested. While *E. carinatus* and *E. c. sochureki* catalysed the hydrolysis of hyaluronic acid with equal efficacy, all three *Bungarus* species exhibited hyaluronidase activities to varying degrees, with *B. caeruleus* exhibiting the highest activity.

**Snake venom protease assay.** Proteases, such as snake venom serine protease and snake venom metalloproteases, are amongst the medically important toxins that are commonly identified in venoms of many snake species [32]. They exert toxicity by targeting plasma proteins involved in the blood coagulation cascade, fibrinolysis, platelet aggregation, haemorrhage, etc [32]. Therefore, in order to understand the proteolytic nature of venoms, we performed colorimetric protease assays using azocaesin as substrate. While the elapid snake venoms of *Naja* and *Bungarus spp*. showed very low or negligible azocaesinolytic activity in relation to the bovine pancreatic protease standard ($p < 0.05$), both *Echis* subspecies exhibited significantly higher activities (Fig 3D). Interestingly, the venom of *E. c. sochureki* showed nearly a two-fold increased activity than its 'big four' counterpart (25% and 43% respectively; $p < 0.05$).

**Fibrinogenolytic assay.** Several components of snake venoms have been shown to affect haemostasis by acting on various factors involved in the blood coagulation pathway [33]. Crude venoms of the medically relevant but neglected snakes and their 'big four' counterparts were tested for fibrinogenolytic activity based on their ability to degrade Aα and Bβ subunits of human fibrinogen. Of all the venoms tested, *Echis* subspecies displayed the highest fibrinogenolytic activity, degrading both Aα and Bβ fibrinogen subunits (S2A–S2C Fig). In time dependent assays, *Echis* venoms completely degraded both Aα and Bβ subunits within the first 15 minutes of incubation (S2D and S2E Fig). With the exception of venom from *B. fasciatus*,

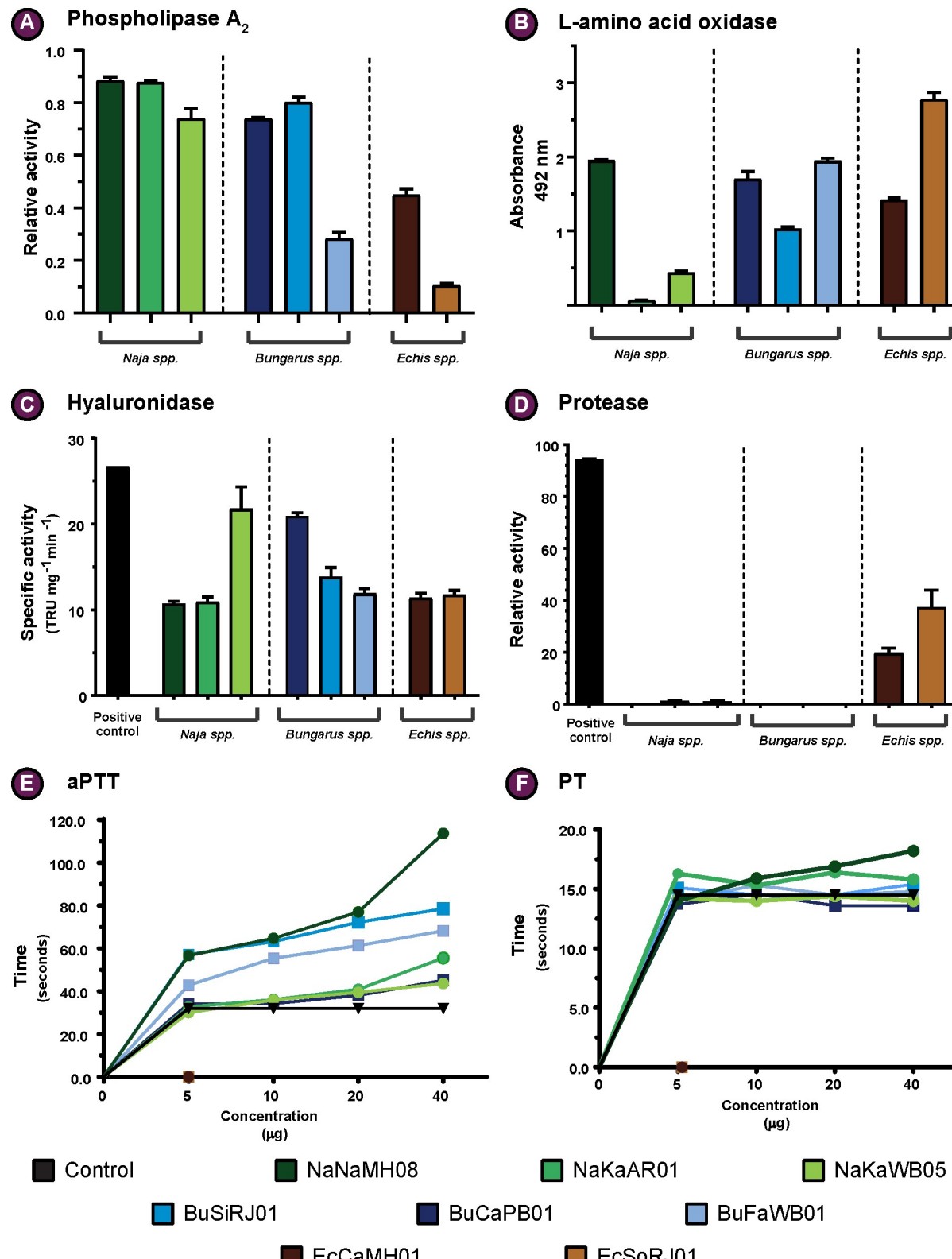

**Fig 3. Distinct biochemical profiles of the neglected snakes and their 'big four' counterparts.** (A) Relative enzymatic activity of PLA$_2$; (B) Relative LAAO activity (absorbance units); (C) Specific hyaluronidase activity (TRU/mg/min); (D) Relative protease activity (%); (E) and (F) dose-dependent effect on activated partial thromboplastin time (aPTT) and prothrombin time (PT), respectively, where time (sec) is plotted

against venom concentration (μg). All assays, with the exception of (E) and (F), were performed in triplicates and the standard deviation is indicated by error bars. A colour code is provided, which corresponds to the respective species of snake.

which failed to exhibit any evidence of fibrinogenolysis, all other elapid species under investigation exhibited Aα- chain specific activity after 1 hour of incubation, in a highly comparable manner.

**DNase assay.** Snake venoms comprise of toxins that cause cell destruction and necrosis, which, in turn, results in the extrusion of DNA and the formation of extracellular traps [34, 35]. In addition to blocking blood vessels, these traps can result in the localized accumulation of tissue destroying toxins [34, 35]. However, DNase, an enzymatic snake venom toxin with endonuclease activity [36, 37], has been shown to prevent the entrapment of these toxins in the extracellular traps [34]. Thus, the presence or absence of DNase might significantly alter clinical manifestations post envenoming. Therefore, we assayed DNase activities in the venoms of both neglected snakes and their 'big four' counterparts. As shown for its 'big four' counterpart [34], the venom of *E. c. sochureki* displayed low to negligible DNase activity (S3A and S3B Fig). On the contrary, the venom of *N. naja* has been shown to exhibit significant DNase activity [34]. Interestingly, while *N. kaouthia* from West Bengal exhibited relatively increased DNase activity than its 'big four' relative, *N. kaouthia* from Arunachal Pradesh displayed very low activity (S3A and S3B Fig). A similar trend was observed in *Bungarus spp*. comparisons, where *B. fasciatus* was found to exhibit maximal DNase activity, while *B. sindanus* showed least activity (S3A and S3B Fig).

**ATPase and ADPase assays.** Although ATPase and ADPase enzymatic toxins are omnipresent in most snake venoms as minor toxin components, their pharmacological roles are not clear yet. The release of purines, mainly adenosine, has been theorized to play a role in inducing an array of clinical manifestations [38]. In this study, *Naja spp*. were found to exhibit profound ATPase and ADPase activities, with the *N. naja* venom exhibiting the highest activity (S4 Fig). In a complete contrast, all three *Bungarus spp*. did not exhibit ATPase or ADPase activities. However, ATPase and ADPase activities of the *E. c. sochureki* venom were comparable to those of the *Naja spp*, while *E. carinatus* exhibited relatively lower activities.

**Blood coagulation assays.** Many snake species are known to have a clinically significant impact on the blood coagulation cascade post envenomation. The venom toxins act on various blood coagulation factors, thereby, affecting blood coagulability, ultimately facilitating other toxins to inflict symptoms such as damaged blood vessels, shock, intracranial and pituitary haemorrhage, renal failure, thrombosis, pulmonary embolism and thrombocytopenia [39–41]. Bearing the significance of these clinical manifestations in mind, we estimated PT and aPTT to understand the effect of venoms under study on the extrinsic and intrinsic blood coagulation pathways, respectively (Fig 3E and 3F; S3A–S3C Table). aPTT measurements revealed that, among *Naja spp*., the venom of *N. naja* from Maharashtra significantly prolonged the blood coagulation time by several seconds in comparison to the control, suggesting its effect on the intrinsic pathway. Similar, anticoagulatory effects were observed in the case of *B. sindanus* and *B. fasciatus* venoms at higher concentrations (20–40 μg). In contrast, the venoms of *Naja* and *Bungarus* species showed only minor deviance from the control values in PT estimation, with INR values being slightly more than 1. This suggests that these venoms do not target the extrinsic or the common pathway of the clotting cascade [42]. On the contrary, blood samples treated with the venoms of *Echis* subspecies clotted instantly, even at the lowest concentration tested (5 μg), during both aPTT and PT tests, highlighting their extremely potent procoagulant effects.

**Haemolytic assay.** PLA$_2$s are amongst snake venom toxins known to cause direct haemolysis of erythrocytes by catalysing the hydrolysis of membrane phospholipids [43, 44]. To

understand the cytotoxic effects of venoms of the neglected species and their closest 'big four' relatives, we conducted *in vitro* haemolytic assays on human red blood cells (S5 Fig). These experiments revealed the maximal haemolytic effects of *N. naja* venom, followed by the two *N. kaouthia* populations. While the venom of *B. caeruleus* was observed to inflict more damage to human RBCs than *B. sindanus* and *B. fasciatus* venoms, the two *Echis* subspecies displayed negligible haemolytic activities.

### Binding efficiency and cross-reactivity of commercial antivenoms against the venoms of neglected species and their 'big four' relatives

The immunological binding of some of the major polyvalent antivenoms marketed in India against 'big four' snake venoms were quantified using endpoint titration ELISA, where various dilutions of commercial antivenoms were incubated with a fixed concentration of venom. The absorbance values at 405 nm, which directly correlate to the binding efficiency of antivenom, were plotted against the dilution factors. Our results revealed that all four marketed antivenoms not only recognized the venoms of the 'big four' representatives (*N. naja*, *B. caeruleus* and *E. carinatus*), but also exhibited varying degrees of cross-reactivity towards the venoms of the neglected species (*N. kaouthia*, *B. sindanus*, *B. fasciatus* and *E. c. sochureki*) (Fig 4). For instance, all four commercial antivenoms tested recognized *N. naja* venom from Maharashtra equally well (titre of 1:2500) but exhibited differential degrees of binding against the venoms of *N. kaouthia* (Fig 4). Both Premium Serums and VINS antivenoms exhibited poor binding against West Bengal and Arunachal Pradesh populations of *N. kaouthia* (titre of 1:500), while Bharat Serums and Haffkine antivenoms exhibited significantly increased binding. Similarly, antivenoms manufactured by Premium Serums, Bharat Serums, and Haffkine Bio-Pharma bound well to the venom of *E. carinatus* (1:2500), while VINS antivenom exhibited poor venom recognition capability (titre of 1:500). In the case of *E. c. sochureki* venom, Premium Serums is the only antivenom which exhibits increased binding (titre of 1:2500), while all other antivenoms failed to recognize the venom effectively (titre 1:500). Most surprisingly, all antivenoms were found to possess poor binding capabilities against the venoms of the three *Bungarus spp.*, including that of *B. caeruleus*, a 'big four' representative whose venom is used in the immunization mixture (an extremely low titre of 1:100 for Premium Serums, and 1:500 for the rest). It should be noted that the estimation of total IgG content in a vial of commercial antivenom revealed that the overall IgG content remains roughly the same across all four manufacturers (6.7 to 9.69 mg/mL; S1B Table). This indicates that the differences observed in absorbance for various antivenoms is suggestive of the actual differences in binding efficiencies, and do not result from the differences in IgG contents.

Further, by employing immunoblotting experiments, we were able to identify several major venom components that are not effectively recognized by the marketed antivenoms (Fig 5A–5F). While all four antivenoms recognized several toxins in the molecular range between 55–70 kDa in venoms of both neglected species and their 'big four' counterparts (as indicated by the intensity of bands), many bands, particularly of higher molecular weight (>70 kDa), remained completely unrecognized (Fig 5C–5F). For example, with the exception of Premium Serums, one of the bands at 55 kDa was not recognized by any of the antivenoms. Similarly, as reported previously in various medically important Indian snakes [45–47], low molecular weight toxins (~10 kDa) were largely unrecognized. Comparison of these immunoblots to the negative control (naive horse IgG) reveals non-specific binding of IgGs to these largely abundant low molecular weight toxins, especially in Elapidae snakes (Fig 5B). Results of immunoblotting experiments largely corroborate findings of indirect ELISAs and highlight the poor venom recognition potential of commercial antivenoms. Interestingly, while ELISA revealed a

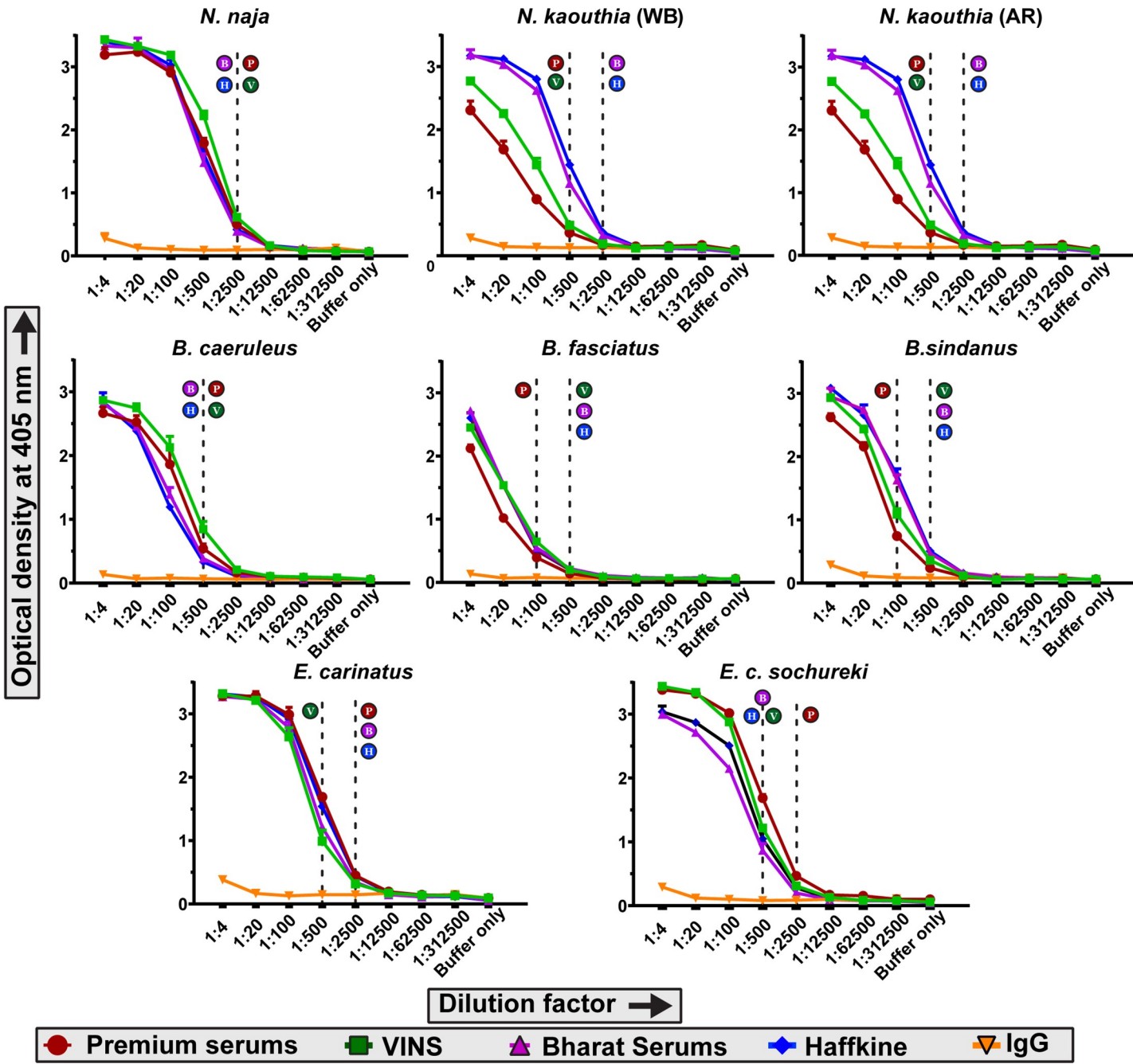

**Fig 4. Binding efficacy of commercial polyvalent antivenoms against the venoms of medically important Indian snakes.** Binding efficacy determined using an end-point ELISA is depicted here. Optical density at 405 nm is plotted against various dilutions of antivenoms. The values are provided as mean absorbance of triplicates, and error bars represent standard deviation. The dotted lines represent antivenom titres, which were determined using purified IgG from unimmunized horses as negative control. Alphabets next to the dotted lines indicate the titre of the respective antivenom (P: Premium Serums & Vaccines Pvt. Ltd.; V: VINS Bioproducts Ltd.; B: Bharat Serums and Vaccines Ltd; and H: Haffkine Institute).

higher titre for Premium Serums antivenom against *E. c. sochureki*, and all antivenoms, with the exception of VINS, against the venom of *E. carinatus*, immunoblotting experiments showed that many toxins in these two subspecies were completely unrecognized (Fig 5C–5F).

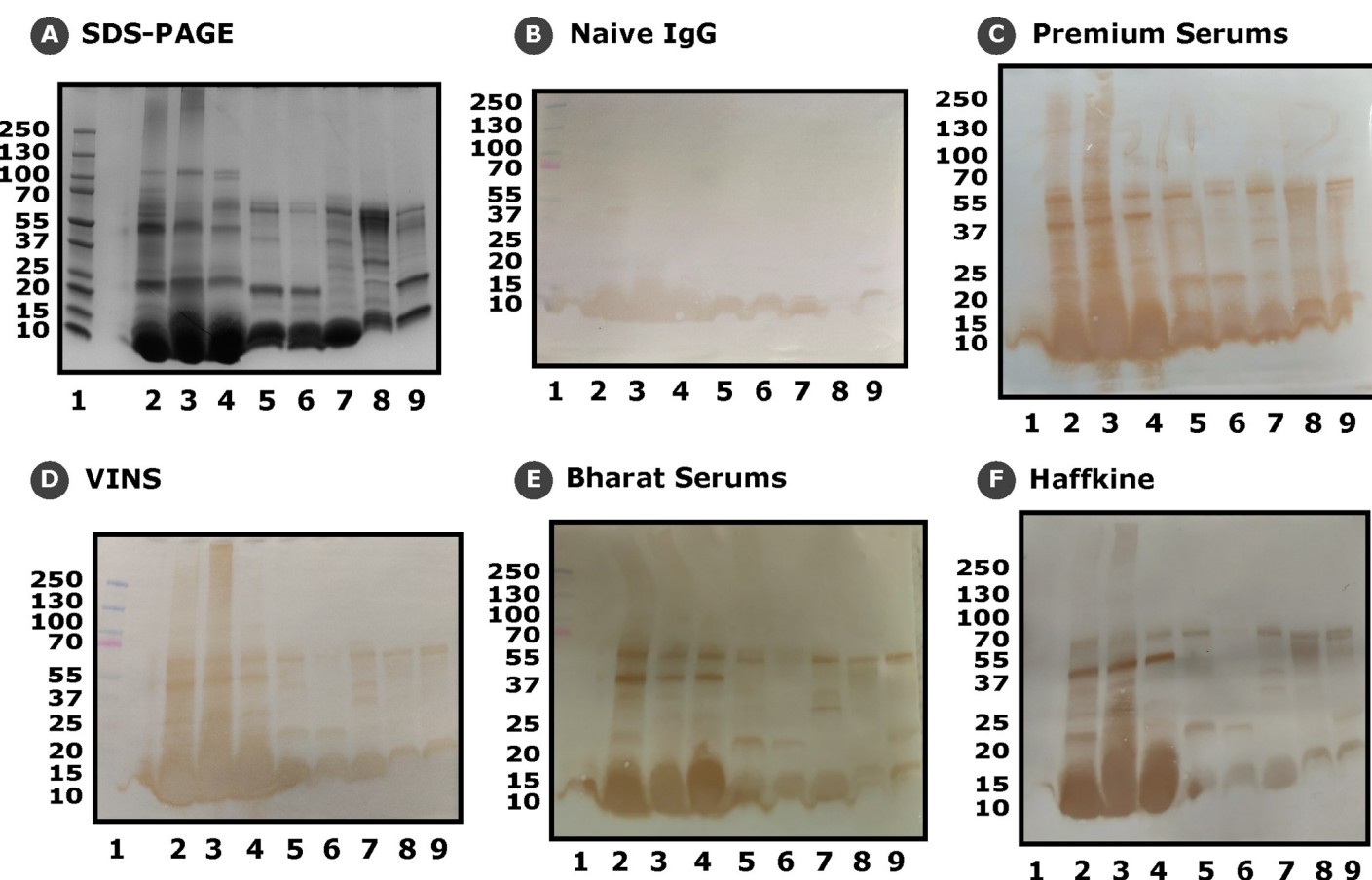

**Fig 5. Identification of untargeted toxins via western blotting.** This figure depicts the results of western blotting experiments. Following the electrophoretic separation of snake venoms (25 μg) on a 4–20% gradient SDS-PAGE (A), bands were electroblotted onto nitrocellulose membranes, and incubated with 1:200 dilutions of various antivenoms: (B) Naïve IgG; (C) Premium Serums & Vaccines Pvt. Ltd.; (D) VINS Bioproducts Ltd.; (E) Bharat Serums; and (F) Haffkine Institute.

## Preclinical venom neutralization

Although these results highlight the poor recognition capabilities of commercial antivenoms against the venoms of both neglected species and their 'big four' counterparts, *in vitro* binding experiments cannot ultimately predict toxin neutralisation do not reveal the underlying neutralization power [16, 48]. Therefore, we conducted WHO-recommended preclinical neutralization assays in mice. In order to determine the neutralization potency, we first estimated the median lethal dose ($LD_{50}$), which is the amount of venom required to kill 50% of the test population, for each snake species under investigation (Fig 6A; S4A Table). These experiments revealed several fascinating aspects about the nature of venom in these medically important snakes. In the mouse model, the venom of *N. kaouthia* from West Bengal (0.24 mg/Kg) was found to be five times more potent than the conspecific population from Arunachal (1.23 mg/Kg) and thrice as potent as its 'big four' counterpart (0.73 mg/Kg). Interestingly, *E. c. sochureki* showed the lowest venom potency amongst all snakes investigated in this study (1.76 mg/Kg). These experiments estimated an $LD_{50}$ of 0.02 mg/Kg for *B. sindanus*, which was five times as potent as *B. caeruleus* (0.1 mg/Kg) and an astonishing 56 times as potent as *B. fasciatus* (1.12 mg/Kg), making it one of the most potent snake venoms in the world against mammals.

Next, we assessed the efficacy of marketed antivenom at neutralizing venom lethality from the neglected species and their 'big four' counterparts by performing $ED_{50}$ experiments (Fig

6B; S4B Table). In order to significantly reduce the number of mice sacrificed in these experiments, we only tested Premium Serums antivenom, which is amongst the most widely marketed polyvalent antivenoms in India. These experiments highlighted the inefficiencies of this commercial Indian antivenom in neutralising bites from several neglected snake species (S4B Table). For example, the neutralising potency against *N. naja* venom (0.717 mg/mL) was as marketed (0.6 mg/mL), however, the neutralising potency against *N. kaouthia* from West Bengal was a meagre 0.156 mg/mL, while it completely failed to neutralize both 5x and 3x $LD_{50}$ of its conspecific population from Arunachal Pradesh. Alarmingly, the neutralising potency of this antivenom against *B. caeruleus*, a venom used in the immunization process of antivenom production, was low (0.312 mg/mL), and failed to meet the marketed label claim (0.45 mg/mL). Similarly, it performed very poorly against *B. sindanus* (0.272 mg/mL), which was found to be the most toxic against mice of all the snakes examined in this study. To our surprise, the antivenom performed better against the venom of *B. fasciatus* with a neutralization potency of 0.643 mg/mL, suggesting that this antivenom may be efficacious at treating bites by this species. The neutralising potency of the Premium Serums antivenom (0.52 mg/mL) met the label claim against the venom of *E. carinatus* (0.45 mg/mL) and exhibited the highest neutralization potency observed when tested against *E. c. sochureki* (1.51 mg/mL), further insinuating that this antivenom could potentially neutralise the lethal effects of the venom of this species.

## Discussion

### Dramatic differences in venom composition and biochemistry of the 'big four' and the 'neglected many'

Research on snake venoms has demonstrated the presence of a wide range of toxin types, capable of exerting variable clinical pathologies following snakebite [49]. The composition and

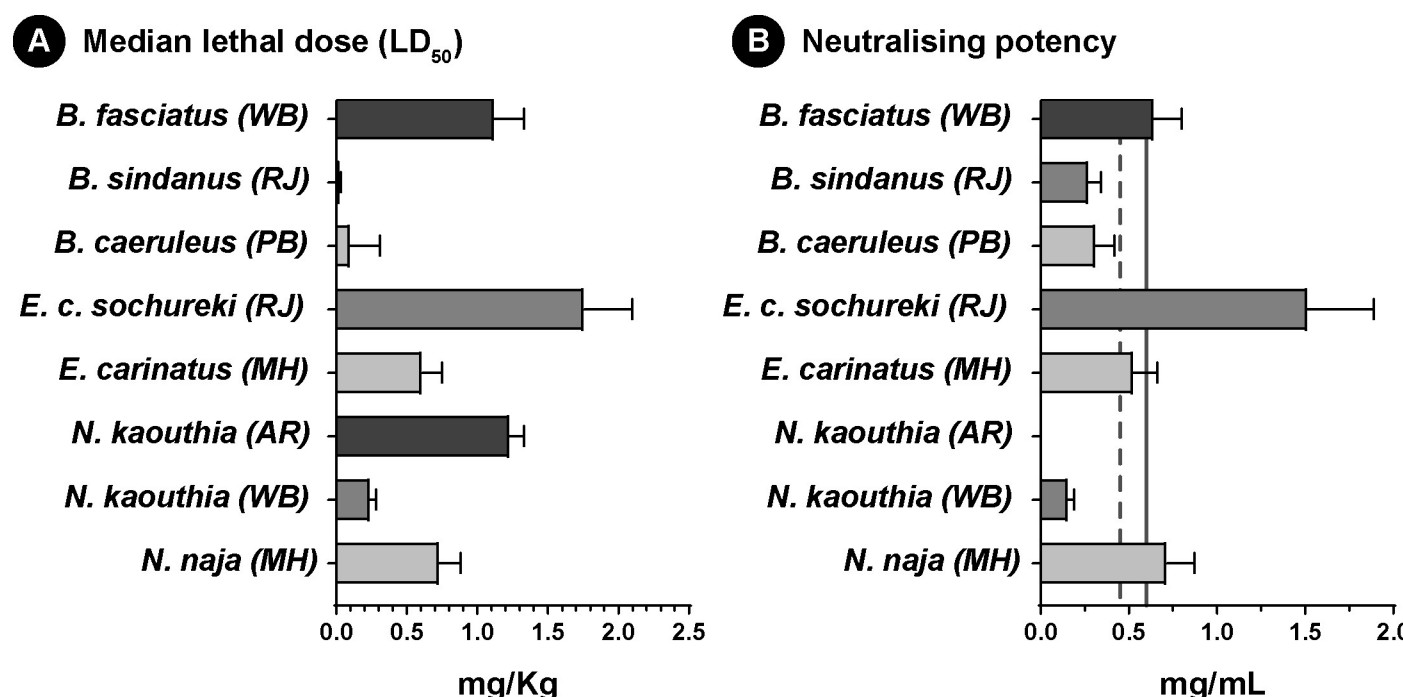

**Fig 6. Toxicity profiles of snake venoms and neutralising potencies of commercial antivenoms.** (A) and (B) depict the median lethal doses of various medically important snakes and the neutralising potency of Premium Serums antivenom, respectively. Vertical solid and dotted lines in panel (B) indicate marketed neutralising potencies of antivenoms against *N. naja* (0.60 mg/mL), and *B. caeruleus* and *E. carinatus* (0.45 mg/mL), respectively.

abundance of venom toxins, which are dictated by prey availability, gender, ontogeny, and environment [50–54], may have profound effects on the resultant clinical manifestations. Therefore, unravelling compositional differences in snake venoms is vital for the mitigation of snakebites [55, 56].

Biochemical and proteomic characterisations of venoms of medically important Indian snakes in this study revealed stark inter- and intraspecific differences in composition and activities. For example, 86% of the venom of *N. kaouthia* (WB) comprised of neurotoxic 3FTx, which was 1.7 and twice the amount found in *N. kaouthia* (AR) and *N. naja*, respectively. This difference could significantly impact prey immobilization, as this subtype of 3FTx targets acetylcholine receptors (nAChR) to exert severe neuroparalytic effects. Unsurprisingly, mice injected with the venom of *N. kaouthia* (WB) population (5 μg) succumbed within 5 mins of injection, while those injected with *N. naja* venom (18 μg) died nearly an hour later. Similarly, nearly five times the amount of cytotoxic 3FTXs detected in *N. kaouthia* (AR) population could result in severe cytotoxic symptoms in their bite victims. Although we detected only 5.4% of cytotoxic 3FTXs in the *N. naja* population from Maharashtra, bites from this population have also been reported to result in severe cytotoxic effects (S6 Fig). Such cytotoxic effects of *Naja* venoms from India have also been described previously [57]. In addition to cytotoxic 3FTXs, PLA$_2$ have also been shown to cause cytotoxicity [58]. Nearly 15% of venom of this *N. naja* population comprised of PLA$_2$s. Perhaps, the combined effect of cytotoxic 3FTXs and PLA$_2$s could be responsible for the necrotic symptoms associated with bites from this population.

PLA$_2$ is another important toxin superfamily that has been documented to exhibit a remarkable diversity of pharmacological effects, including neurotoxicity, myotoxicity, haemotoxicity, initiation/inhibition of platelet aggregation, and anticoagulation [59]. PLA$_2$ assays revealed drastic differences in phospholipase activities in venoms of the neglected snakes and their 'big four' counterparts (Fig 3A; S1 Fig). Efficiency in catalysing the breakdown of phospholipids, which can potentially contribute to toxicity [25], varied dramatically across species, with the *Naja spp.* exhibiting the highest activity despite having significantly lower PLA$_2$ content in comparison to the two *Echis* subspecies. The result is not surprising as most Viperidae snake venom PLA$_2$s are known to contain a replacement substitution at the catalytic site (49$^{th}$ residue) that diminishes their phospholipase activity [60]. In case of *Bungarus spp.*, where *B. caeruleus* and *B. sindanus* displayed significantly high and dose-dependent increase in PLA$_2$ activity in comparison to *B. fasciatus* (Fig 3A; S1 Fig), even though they contained two to three times less PLA$_2$ than the latter species. Although these differences, especially at lower venom concentrations, may seem negligible, even minor differences in biochemical profiles may impose profound biological implications, particularly in snakes that inject relatively lower quantities of venom, including many species of kraits and saw-scaled vipers.

Snake venom toxins are also known to perturb haemostasis by targeting either the blood coagulation factors, platelets, red blood cells or the extracellular matrix of blood vessels [61]. Evolution of divergent mechanisms in Elapidae snakes for inducing fibrinogenolysis has been previously attributed to result in anticoagulation [62, 63]. The venoms of *Naja* and *Bungarus spp.* tested in our study displayed fibrinogenolytic activity and significantly delayed the coagulation time of the human plasma, thereby, highlighting their anticoagulatory nature (Fig 3E; S2A–S2C Fig; S3A–S3C Table). PLA$_2$s can also interfere with haemostasis by not only preventing clot formation via hydrolysis of procoagulant phospholipids [61], but also by causing haemolysis by disrupting erythrocyte membranes [61, 64] and by inhibiting platelet aggregation [65, 66]. In alignment with the determined catalytic efficiencies, crude venoms of *Naja* and *Bungarus spp.* displayed increased haemolytic and anticoagulant activities (Fig 3E; S3A–S3C Table; S5 Fig). The latter is achieved in both elapid species via the disruption of the intrinsic

coagulation cascade (Fig 3E and 3F; S3A–S3C Table). Venom ADPases are also known to contribute to the anticoagulant property of some snakes [67, 68]. This enzymatic toxin cleaves ADP, thereby, interferes with the platelet aggregation mechanism of blood coagulation [69]. Interestingly, the venoms of *Naja spp*. exhibited higher catalytic cleavage of ADP substrate (S4 Fig), insinuating that these snakes employ several toxins that synergistically disrupt haemostasis.

Proteomic characterisation further led to the identification of several toxins in *Echis* venoms that may perturb the blood coagulation cascade of the bite victim. For example, SVSP is an enzymatic toxin that targets various components of the coagulation pathway to induce haemorrhage, apoptosis, fibrinogenolysis, and inhibition of platelet aggregation [70–73]. We identified 3–7.4% of SVSPs in the two *Echis* subspecies and detected appreciable proteolytic activity. *Echis* venoms have been previously shown to induce blood coagulation via prothrombin activating PIII SVMPs [74]. Furthermore, in our experiments, *Echis* venoms also displayed tremendous fibrinogenolytic activity, cleaving both Aα and Bβ subunits of the human fibrinogen in under 5 mins, which is also likely to contribute to their extreme procoagulant effects (S2 Fig) [75, 76]. Similarly, LAAO and snaclec are also known to affect haemostasis by causing impairment of platelet aggregation and thrombocytopenia, respectively [77, 78]. Although LAAO was detected to comprise a large proportion of *E. carinatus* venom (25.1%), in comparison to *E. c. sochureki* venom (12.7%), the latter species exhibited relatively increased LAAO activity.

While the mechanism of action of some of the snake venom toxins have been extensively studied [70, 79, 80], others, especially the least abundant toxins, remain chiefly underinvestigated. For example, CRISPs are known to exhibit a wide range of toxic effects in the venoms of Toxicofera reptiles via the activation or inhibition of various physiologically important ion channels [59]. Therefore, the proteomic differences observed in the relative amounts of CRISP toxins are likely to have a profound effect on the clinical profile, whereas the differences in the relative abundance of CVF, a complement-activating protein, mostly abundant in cobra venoms (Fig 2), whose precise function is yet to be elucidated, may not result in a clinically significant difference [81]. Snake venom toxins have also been theorised to indirectly exert toxicity by functioning in concert with other venom proteins. Hyaluronidase, a lowly secreted toxin in most snake venoms, represents a classical example of synergistically functioning venom proteins [82]. It has been shown to degrade hyaluronic acid in the extracellular matrix, causing permanent damage to the local tissues, while also promoting the 'spread' of other venom components by reducing the viscosity and increasing the permeability of cellular membranes [22, 82]. Therefore, despite being expressed in limited quantities, hyaluronidase has a greater clinical significance. The proteomic characterisation of venoms under study detected very negligible quantities of hyaluronidase in *Naja* and *Bungarus spp*. (<1%), in comparison to *Echis* (1.7 to ~4%) and, yet, the catalytic degradation of hyaluronic acid by *N. kaouthia* (WB) and *B. caeruleus* was higher than all the other venoms tested, and was comparable to the purified sheep testis hyaluronidase (positive control), highlighting the phenomenal difference in the efficiencies of these components.

Thus, we highlight a remarkable complexity in proteomic compositions and biochemistries of neglected species and their 'big four' counterparts, which, in turn, may affect the clinical outcome of bites from these snakes.

## Stark differences in venom potencies may be reflective of variation in feeding ecologies

The evolution of venom has underpinned the successful diversification of venomous lineages across the animal kingdom by enhancing their predatory and/or defensive abilities [83, 84].

Hence, the composition of venom has been proposed to be predominantly dictated by the spectrum of prey and predators encountered [85, 86]. Remarkable specificity of snake venoms towards their natural prey has been previously demonstrated. For instance, the venom of *E. c. sochureki* from Pakistan, was earlier shown to exhibit increased specificity towards scorpions in comparison to mice [87]. Similarly, Northwest Indian population of *E. c. sochureki* may have also specialized on non-mammalian prey and, hence, their venom was found to exhibit least toxicity to mammals. Results of lethality experiments (mouse model) in this study corroborate these predictions, and *E. c. sochureki* was found to be the least potent of all snakes investigated (five times less potent than its 'big four' counterpart from Maharashtra) (Fig 6A; S4A Table). Interesting differences in toxicities were also observed in *Bungarus spp*. Though the dietary preferences of *B. sindanus* remains unclear, the population from North West India proved to be the most potent snake (in mice model), in comparison to *B. caeruleus* from the Indian subcontinent [88, 89] (Fig 6A; S4A Table). Extreme toxicity towards mice with an $LD_{50}$ of 0.02 mg/Kg–which, perhaps, makes them the most toxic snakes in India, suggests that they may predominantly prey on mammals. On the other hand, *B. fasciatus*, which is known to chiefly prey on snakes and other reptiles [90, 91], exhibited decreased toxicity towards mice (11 and 56 times lower than *B. caeruleus* and *B. sindanus*, respectively).

The composition of venom and associated toxicity is also theorized to vary across geographical regions [92, 93], as the availability of prey is dependent on the biotic and abiotic factors of the region. Toxicity profiles of *N. kaouthia* populations from East and Northeast India exhibited significant intraspecific differences, wherein, the former population was found to be extremely potent (0.24 mg/Kg), killing the experimental animals within minutes of venom administration, while the latter (1.23 mg/Kg) took several hours (Fig 6A; S4A Table). Additionally, the population of *N. kaouthia* from Arunachal Pradesh also exhibited lower toxicity in comparison to the other Northeast Indian and Bangladesh populations investigated previously [45, 94].

## Limited venom binding and neutralisation by the 'big four' antivenom warrants the development of specific and effective snakebite therapy for the 'neglected many'

Despite the frequent incidence of potentially lethal bites from the neglected species of snakes under investigation, specific antivenoms for the treatment of envenomation by these species do not exist. For the treatment of bites in regions where these species are of medical concern, the polyvalent 'big four' antivenom is currently employed by the clinicians. Therefore, to evaluate the effectiveness of currently marketed 'big four' antivenoms in neutralising bites from these neglected species, we carried out WHO-recommended *in vitro* and *in vivo* preclinical evaluations. Our *in vitro* results demonstrate that the commercial antivenoms are poor at recognising the venoms of *B. sindanus*, *B. fasciatus*, *E. c. sochureki*, and the two *N. kaouthia* populations (Fig 4). For instance, two of the four antivenoms tested (Premium Serums and VINS) recognised *N. naja* venom from Maharashtra—which is used in the immunisation mixture, two to five times more effectively than the venoms of *N. kaouthia* from the eastern- and north-eastern India. Similarly, with the exception of VINS (titre of 1:500), all other antivenoms recognized *E. carinatus* venom effectively (titre of 1:2500), while binding fairly poorly to the venom of *E. c. sochureki* (titre of 1:500). Only Premium Serums antivenom displayed significant cross reactivity against the venom of the Sochurek's viper and recognized the venom at even very high dilutions (titre of 1:2500). Surprisingly, all four commercially marketed antivenoms not only poorly recognized the venom of *B. sindanus* and *B. fasciatus* but also that of *B. caeruleus*—a 'big four' representative (titre of 1:500)

(Fig 4). The immunoblotting results painted a similar picture where the antivenoms exhibited cross-reactivity towards a few toxins of *N. kaouthia* and *E. c. sochureki*, while unfortunately, a vast majority of toxins of the neglected species remained unrecognized (Fig 5). This suggests that while commercial antivenoms may have some paraspecific IgGs that cross react with certain toxins found in the venoms of the neglected species, many are not recognised, thereby, emphasising severe clinical limitation of the current snakebite therapy. Moreover, the apparent recognition of low molecular weight toxins by commercial antivenoms was revealed by the complete non-specific binding exhibited by the purified IgG from naïve horses (Fig 5B). Thus, our *in vitro* immunological experiments highlight the poor recognition capabilities of the commercial Indian antivenoms against the venoms of the neglected species, as well as against one of the 'big four' snake venoms that is used in the antivenom manufacturing process.

Furthermore, the results of our *in vitro* assays strongly corroborate the findings of *in vivo* neutralisation experiments. Although the majority of antivenom manufacturers in the country primarily source venoms from the Irula Snake Catchers Industrial Cooperative Society in Tamil Nadu (South India) for antivenom production, the tested antivenoms showed strong binding to the western population of *N. naja*. Moreover, the polyvalent antivenom marketed by Premium serums exhibited the advertised neutralising potency against this population (Fig 6B; S4 Table). This could be attributed to the infrequent supplement of venoms sourced from Maharashtra into the immunization mixture (KS and RW's personal communication with Indian antivenom manufacturers), as well as cross neutralising capabilities of antivenoms. On the other hand, the polyvalent antivenom failed to effectively neutralize the lethal effects of both populations of *N. kaouthia* in the mouse model. While the neutralisation potency of the antivenom was poor against the East Indian population, the polyvalent antivenom completely failed to neutralize even 3x $LD_{50}$ of *N. kaouthia* venom from Northeast India, despite this population being the least toxic among the *Naja* venoms tested (Fig 6B; S4B Table). In the case of *Echis*, venoms of both subspecies were effectively neutralized by the polyvalent antivenom. This result for *E. carinatus* is not surprising as its venom is included in the immunization mixture for the production of antivenom. However, the surprisingly effective neutralization of *E. c. sochureki* venom could be due to the presence of paraspecific neutralising antibodies (Fig 6B; S4B Table). Since $ED_{50}$ experiments are not reflective of the capabilities of antivenoms in preventing morbid symptoms inflicted by snake venoms, further preclinical investigation, particularly in the case of *E. c. sochureki*, is warranted. As predicted by the *in vitro* assays, the antivenom could not effectively neutralize neither the venom of *B. caeruleus* from North India (Punjab), nor *B. sindanus* from West India (Rajasthan) (Fig 6B; S4B Table). These results contrasted the reports of effective neutralization of *B. caeruleus* venom from Sri Lanka and South India [88, 89], as well as moderate neutralisation of venoms of *B. caeruleus* and *B. sindanus* from Pakistan [88, 89] by Indian polyvalent antivenoms. However, our results support previous findings where the marketed antivenom was shown to be inefficient in neutralising the venom of *B. sindanus* from Pakistan in *in vitro* neurotoxicity assays [95]. Thus, these results highlight the poor neutralising capabilities of marketed antivenoms and emphasize the need for the production of effective antivenoms against the 'neglected many'. However, due to ethical considerations, neutralisation assays were performed only for one of the commercially marketed Indian antivenoms. The effectiveness of the other antivenoms in neutralising bites from the neglected species requires further validation through preclinical and clinical research [96, 97]. Since none of these antivenom manufacturers utilize the venoms of the neglected species in the immunisation mixture, we do not expect them to perform any better than the antivenom tested in this study.

## Future directions for the production of highly efficacious paraspecific Indian antivenoms

Historically, snakebite has been a life-threatening tropical disease, mostly affecting the agrarian and rural communities of India. Despite technological advancements, especially in the area of recombinant technology, antivenoms are manufactured using a century old strategy. The conventional animal derived antivenoms, which may possess impurities and IgGs not specific to venom toxins or medically important toxins, increases non-specificity and immunogenicity, leading to anaphylaxis [98]. Moreover, the commercial antivenoms are only manufactured against the four widely distributed medically important species, popularly known as the 'big four' snakes. However, numerous other geographically restricted snake species, which contribute to snakebite morbidity and fatalities, have remained neglected, and their bites are treated with the antivenom raised against the 'big four' snakes. For example, various medically important species of cobras (*N. sagittifera*, *N. oxiana*, *N. kaouthia*), kraits (*B. andamanensis*, *B. fasciatus*, *B. niger*, *B. sindanus*), vipers (*Hypnale hypnale*, *Ovophis monticola*, *E. c. sochureki*, *Macrovipera labetina*), coral snakes (*Calliophis nigrescens*, *Sinomicrurus macclellandi*), sea snakes (*Pelamis platurus*, *Enhydrina schistose*, *Hydrophis cyanocinctus*) and sea kraits (*Laticauda colubrina*), etc., are capable of delivering clinically significant and, even, fatal bites but specific antivenoms for treatment of envenomation by these snakes do not exist. Results of *in vitro* and *in vivo* experiments in this study highlight the inefficiencies of the Indian polyvalent antivenoms in recognising and neutralising the venoms of some of these neglected snakes, including *B. sindanus* and the East and Northeast Indian populations of *N. kaouthia* (Figs 4, 5 and 6B; S4B Table). Therefore, the development of specific antivenoms—either monovalent or broadly neutralising polyvalent in nature, effective against these neglected snakes, is the need of the hour. Since the majority of bite victims do not see the snake, and commoners and clinicians lack training in identifying snake species, the development of a rapid, efficient and accurate snakebite diagnostics kit becomes indispensable. The lack of such a technology in India rules out the utilization of monovalent antivenoms. Hence, the optimal strategy for producing antivenoms against the 'neglected many' in regions where they are prevalent involves the inclusion of their venoms in the immunization mixture.

Estimation of protein content of the marketed Indian antivenoms revealed that they contain as little as 6 to 9 mg/mL of proteins/IgG (S1B Table). In comparison, antivenoms produced for treating bites from African snakes and certain sea snake species have been documented to contain 10 to 20 times more protein/IgG, respectively [48, 99]. The reduced concentration of venom specific IgGs in Indian antivenoms almost certainly results in increased number of vials required to effect cure, which, in turn, not only increases the cost of treatment but also the risk to developing serum sickness or anaphylaxis as a result of administration of large amounts of non-specific IgGs in the bite victim. However, to justify the reduced IgG content of antivenom vials, Indian antivenom manufacturers often cite the WHO manual, which says that the total protein content in antivenom vials should not exceed 10% [17, 100]. The WHO manual further adds that nearly 90% of this protein content should consist of antibodies specific to snake venom toxins. Unfortunately, however, as demonstrated by our *in vitro* and *in vivo* experiments, the current effectiveness of Indian antivenoms is extremely poor, not only towards the neglected species (e.g., *N. Kaouthia* and *B. sindanus*) but also one of the 'big four' representatives, the North Indian population of *B. caeruleus*. In addition to geographic variability in venoms [92, 93], the lack of adherence of Indian antivenom manufacturers to international production standards may also result in these undesirable outcomes.

Hence, there is an urgent need for the introduction of highly specific, dose-effective polyvalent antivenoms that can broadly neutralise bites from the 'big four' and other medically

important yet neglected snakes of the region where the product is marketed. Additionally, the commercialisation of this life-saving drug, which is currently decided by a tender system in various Indian States, should rather be governed by the outcomes of stringent preclinical evaluations. These measures, being implemented along with the improvements in the manufacturing strategy of antivenoms, holds the potential to significantly save lives, limbs, and livelihoods of thousands of snakebite victims and their families in India.

## Supporting information

**S1 Fig. Phospholipase A2 (PLA$_2$) enzyme kinetics.** Kinetic activities of venom PLA$_2$s of the medically important *Naja spp.*, *Bungarus spp.* and *Echis* subspecies at 0.01 μg, 0.1 μg, 0.5 μg and 1 μg of venom concentrations are shown here. The change in the activity of venom PLA$_2$s with time is measured by plotting the absorbance (OD) at 740 nm of the phospholipid substrate every minute over a time interval of 1 hour. The assay was performed in triplicates and the standard deviation is indicated by the error bars.
(PDF)

**S2 Fig. Fibrinogenolytic activities of medically important snake venoms.** This figure shows fibrinogenolytic activities of (A) *Naja spp.*, (B) *Bungarus spp.*, and (C) *Echis* subspecies, and the time-dependent fibrinogenolytic activities of (D) *E. carinatus* and (E) *E. c. sochureki* venoms. M: Pre-stained protein ladder; HF: human fibrinogen; Nn: *N. naja*; Nk: *N. kaouthia* (AR); *N. kaouthia* (WB); Bc: *B. caeruleus*; Bs: *B. sindanus*; Bf: *B. fasciatus*; Ec: *E. carinatus*; Es: *E. c. sochureki* venoms; 1' - 60': Time lapsed in minutes.
(PDF)

**S3 Fig. DNase activities of *Naja*, *Bungarus*, and *Echis* venoms.** (A) Horizontal electrophoresis showing DNase activities. Lane 1: DNA (negative control); 2: DNA + 15 U DNase (positive control); 3: DNA + 50 μg/mL of NaNaMH08; 4: DNA + 50 μg/mL of NaKaAR01; 5: DNA + 50 μg/mL of NaKaWB05; 6: DNA (negative control); 7: DNA + 15 U DNase (positive control); 8: DNA + 50 μg/mL of BuCaPB01; 9: DNA + 50 μg/mL of BuSiRJ01; 10: DNA + 50 μg/mL of BuFaWB01; 11: DNA (negative control); 12: DNA + 15 U DNase (positive control); 13: DNA + 50 μg/mL of EcCaMH01; 14: DNA + 50 μg/mL of EcSoRJ01; (B) Relative DNase activities of *Naja*, *Bungarus*, and *Echis* venoms.
(PDF)

**S4 Fig. ATPase and ADPase activities of *Naja*, *Bungarus*, and *Echis* venoms.** Assays were performed in triplicates, and the absorbance was measured at 820 nm after stopping the reactions at 1 and 3 hours. The standard deviation is indicated by the error bars.
(PDF)

**S5 Fig. Dose-dependent haemolytic activities of neglected snakes and their 'big four' counterparts.** This figure depicts dose-dependent haemolytic activities of the venoms under study against 1% human erythrocyte solution. The activity was estimated based on absorbance at 545 nm. The assays were performed in triplicates and the standard deviation is indicated by the error bars.
(PDF)

**S6 Fig. Cytotoxic effects of *N. naja* envenomations in Maharashtra.** Pictures below of a four-year-old boy, bitten at the lateral side of the right foot, depict the cytotoxic effects associated with *N. naja* envenomations in West India (Maharashtra State). Despite the timely administration of antivenom, local necrosis was observed four to five days post envenomation, highlighting the inefficacy of commercial antivenoms in neutralizing cytotoxic symptoms

caused by this *Naja* population. Interestingly, the proteomic characterization in this study revealed that only 5.4% of venom from this population of *N. naja* was found to comprise of cytotoxic 3FTxs. Photo and information credits: Dr. Sadananda Raut, Dr. Minoo Mehata memorial hospital, Narayangaon, Pune, Maharashtra.
(PDF)

**S1 Table.** A-B. Details of the snake venom and anti-snake venom tested.
(PDF)

**S2 Table. A-H. Components identified via tandem mass spectrometry of medically important snake venoms.** For the identification of toxin classes present in the venom, PEAKS Studio X was used to search the raw MS/MS spectra against Uniprot's SwissProt database (December 2018; www.uniprot.org), and the key statistics of these searches have been shown below. These include, identified protein groups, number of high-confidence peptides and unique peptides (mapping to only one group) supporting these protein groups, percentage of the protein sequence covered by supporting peptides (coverage), the area under the curve of the peptide feature found at the same m/z and retention time as the MS/MS scan (area), and the average molecular mass in KDa. Accession number, the name of species, and the toxin family of the matching uniport entry have also been provided. The percentage, indicated adjacent to the toxin family, corresponds to its proportion in the venoms of (A) *N. naja* from Maharashtra; (B) *N. kaouthia* from Arunachal Pradesh; (C) *N. kaouthia* from West Bengal; (D) *E. carinatus* from Maharashtra; (E) *E. c. sochureki* from Rajasthan; (F) *B. caeruleus* from Punjab; (G) *B. sindanus* from Rajasthan; (H) *B. fasciatus* from West Bengal, as determined by tandem mass spectrometry.
(PDF)

**S3 Table. Coagulopathy associated with envenomation by *Naja*, *Bungarus*, and *Echis* venoms.** The following tables provide dose dependent effects of (A) *Naja spp.*, (B) *Bungarus spp.*, and (C) *Echis* subspecies venoms on extrinsic [Prothrombin time test (PT) and International Normalized Ratio (INR)] and intrinsic [Activated Partial Thromboplastin Time test (aPTT)] blood coagulation pathways. The delay in clotting time (or the time taken for the formation of the first fibrin strands), relative to the control sample in each test, is indicated by a color gradient from red to blue. *Blood clots immediately.
(PDF)

**S4 Table. A. Median lethal dose (LD$_{50}$) of medically important snakes.** This table provides LD$_{50}$ values (in μg/mouse and mg/Kg) of various neglected snakes and their 'big four' counterparts. **B. Median effective dose (ED$_{50}$) of Premium serums antivenom.** This table provides ED$_{50}$ values and, estimated and marketed neutralizing potencies of the tested commercial antivenom (Premium Serums & Vaccines Pvt. Ltd.) against various species of medically important Indian snakes. For species, where the estimated neutralising potency meets the marketed potency of the commercial antivenom or that of its 'big four' counterpart [e.g., *N. naja* (0.60 mg/mL), and *B. caeruleus* and *E. carinatus* (0.45 mg/mL)], the cells are indicated in green. Venoms that are not neutralized by the antivenom in the mouse challenge model are indicated in light red. *antivenom failed to neutralize 5x and 3x LD$_{50}$ venom doses.
(PDF)

## Acknowledgments

This work is part of the Scientific Research Partnership for Neglected Tropical Snakebite (SRPNTS) consortium. Authors are indebted to Drs. Nicholas Casewell, Robert A. Harrison,

Jaffer Alsolaiss, and Stuart Ainsworth (Liverpool School of Tropical Medicine, UK) for providing invaluable inputs. Authors are thankful to Prof. Utpal Tatu, Chinmay Narayana, and Darshak Gadara (Indian Institute of Science, Bangalore) for assistance with tandem mass spectrometry, and to the following State Forest Departments for the kind support and permits for venom collection: Maharashtra, Rajasthan, Arunachal Pradesh, West Bengal, and Punjab. For the invaluable assistance in the collection of samples, authors are thankful to: Sumanth Madhav (Humane Society International), Ajay Kartik and Allwin Jesudasan (Madras Crocodile Bank Trust & Centre for Herpetology), and P. Gowri Shankar (North Orissa University). For contributing photographs in Fig 1, the authors are thankful to Chaitanya Shukla (*B. sindanus* and *B. fasciatus*), Vishal Santra (*N. kaouthia*, West Bengal) and GM (all other photographs). The authors are also thankful to Dr. Sadananda Raut for providing insights and information on the clinical implications of *N. naja* bites from Maharashtra.

## Author Contributions

**Conceptualization:** Kartik Sunagar.

**Formal analysis:** R. R. Senji Laxme, Suyog Khochare, Hugo Francisco de Souza, Bharat Ahuja, Vivek Suranse, Kartik Sunagar.

**Funding acquisition:** Kartik Sunagar.

**Investigation:** R. R. Senji Laxme, Suyog Khochare, Vivek Suranse, Kartik Sunagar.

**Methodology:** R. R. Senji Laxme, Kartik Sunagar.

**Project administration:** Kartik Sunagar.

**Resources:** Gerard Martin, Romulus Whitaker, Kartik Sunagar.

**Supervision:** Kartik Sunagar.

**Visualization:** Kartik Sunagar.

**Writing – original draft:** R. R. Senji Laxme, Kartik Sunagar.

**Writing – review & editing:** R. R. Senji Laxme, Suyog Khochare, Romulus Whitaker, Kartik Sunagar.

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
