## [Decision Letter · Decision Letter 0]

26 Sep 2019

Dear Dr Sunagar:

Thank you very much for submitting your manuscript "Beyond the big four: Venom profiling of the medically important yet neglected Indian snakes reveals disturbing antivenom deficiencies" (#PNTD-D-19-01352) for review by PLOS Neglected Tropical Diseases. Your manuscript was fully evaluated at the editorial level and by independent peer reviewers. The reviewers appreciated the attention to an important problem, but raised some substantial concerns about the manuscript as it currently stands. These issues must be addressed before we would be willing to consider a revised version of your study. We cannot, of course, promise publication at that time.

We therefore ask you to modify the manuscript according to the review recommendations before we can consider your manuscript for acceptance. Your revisions should address the specific points made by each reviewer. 

When you are ready to resubmit, please be prepared to upload the following:

(1) A letter containing a detailed list of your responses to the review comments and a description of the changes you have made in the manuscript.

(2) Two versions of the manuscript: one with either highlights or tracked changes denoting where the text has been changed (uploaded as a "Revised Article with Changes Highlighted" file); the other a clean version (uploaded as the article file).

(3) If available, a striking still image (a new image if one is available or an existing one from within your manuscript). If your manuscript is accepted for publication, this image may be featured on our website. Images should ideally be high resolution, eye-catching, single panel images; where one is available, please use 'add file' at the time of resubmission and select 'striking image' as the file type. 

Please provide a short caption, including credits, uploaded as a separate "Other" file. If your image is from someone other than yourself, please ensure that the artist has read and agreed to the terms and conditions of the Creative Commons Attribution License at http://journals.plos.org/plosntds/s/content-license (NOTE: we cannot publish copyrighted images). 

(4) If applicable, we encourage you to add a list of accession numbers/ID numbers for genes and proteins mentioned in the text (these should be listed as a paragraph at the end of the manuscript). You can supply accession numbers for any database, so long as the database is publicly accessible and stable. Examples include LocusLink and SwissProt.

(5) To enhance the reproducibility of your results, we recommend that you deposit your laboratory protocols in protocols.io, where a protocol can be assigned its own identifier (DOI) such that it can be cited independently in the future. For instructions see http://journals.plos.org/plosntds/s/submission-guidelines#loc-methods

While revising your submission, please upload your figure files to the Preflight Analysis and Conversion Engine (PACE) digital diagnostic tool, https://pacev2.apexcovantage.com/ PACE helps ensure that figures meet PLOS requirements. To use PACE, you must first register as a user. Then, login and navigate to the UPLOAD tab, where you will find detailed instructions on how to use the tool. If you encounter any issues or have any questions when using PACE, please email us at figures@plos.org.

We hope to receive your revised manuscript by Nov 25 2019 11:59PM. If you anticipate any delay in its return, we ask that you let us know the expected resubmission date by replying to this email.

To submit a revision, go to https://www.editorialmanager.com/pntd/ and log in as an Author. You will see a menu item call Submission Needing Revision. You will find your submission record there. 

Sincerely,

Philippe BILLIALD

Associate Editor

José Gutiérrez

Deputy Editor

Reviewer's Responses to Questions

**Key Review Criteria Required for Acceptance?**

**Methods**

-Are the objectives of the study clearly articulated with a clear testable hypothesis stated?

-Is the study design appropriate to address the stated objectives?

-Is the population clearly described and appropriate for the hypothesis being tested?

-Is the sample size sufficient to ensure adequate power to address the hypothesis being tested?

-Were correct statistical analysis used to support conclusions?

-Are there concerns about ethical or regulatory requirements being met?

Reviewer #1: It appears that in their coagulation assays, the authors did not include calcium. This is critical as citrating plasma strips out the calcium, which must be added back in to restore physiological conditions. Otherwise this will dramatically skew the results. This has been shown previously to be a significant consideration with Echis venoms, which vary tremendously in their reliance on calcium and therefore the venoms will be differentially affected as the two species differ 4x in their relative reliance on calcium, and therefore the perceived potency significantly erroneous. Phospholipid dependency is another significant methodological shortcoming. As per: Differential procoagulant effects of saw-scaled viper (Serpentes: Viperidae: Echis) snake venoms on human plasma and the narrow taxonomic ranges of antivenom efficacies, as per: Rogalski A, Soerensen C, Op den Brouw B, Lister C, Dashevsky D, Arbuckle K, Gloria A, Zdenek CN, Casewell NR, Gutiérrez JM, Wüster W, Ali SA, Masci P, Rowley P, Frank N, Fry BG.

Toxicol Lett. 2017 Oct 5;280:159-170

Reviewer #2: (No Response)

**Results**

-Does the analysis presented match the analysis plan?

-Are the results clearly and completely presented?

-Are the figures (Tables, Images) of sufficient quality for clarity?

Reviewer #1: The results are well presented other than being erroneous in the coagulation analysing protocol.

Reviewer #2: (No Response)

**Conclusions**

-Are the conclusions supported by the data presented?

-Are the limitations of analysis clearly described?

-Do the authors discuss how these data can be helpful to advance our understanding of the topic under study?

-Is public health relevance addressed?

Reviewer #1: The conclusions in regards to the two Echis are skewed due to the methological issues, which will be remedied by including calcium and phospholipid in the protocol setup.

Also, the Echis procoagulant activity is well characterised as being driven by prothrombin activating P-III SVMP

Reviewer #2: (No Response)

**Editorial and Data Presentation Modifications?**

Reviewer #1: OK

Reviewer #2: (No Response)

**Summary and General Comments**

Reviewer #1: Prior work that should be cited are:

Cytotoxic effects of Naja venoms

Panagides N, Jackson TN, Ikonomopoulou MP, Arbuckle K, Pretzler R, Yang DC, Ali SA, Koludarov I, Dobson J, Sanker B, Asselin A, Santana RC, Hendrikx I, van der Ploeg H, Tai-A-Pin J, van den Bergh R, Kerkkamp HM, Vonk FJ, Naude A, Strydom MA, Jacobsz L, Dunstan N, Jaeger M, Hodgson WC, Miles J, Fry BG. (2017) How the cobra got its flesh-eating venom: cytotoxicity as a defensive innovation and its co-evolution with hooding, aposematic marking, and spitting. Toxins (Basel). 9(3). pii: E103. doi: 10.3390/toxins9030103.

Coagulation effects of Naja venoms

Differential destructive (non-clotting) fibrinogenolytic activity in Afro-Asian elapid snake venoms and the links to defensive hooding behavior. Mátyás A. Bittenbinder, James S. Dobson, Christina N. Zdenek, Bianca op den Brouw, Arno Naude, Freek J. Vonk, Bryan G. Fry Toxicology in Vitro 60 (2019) 330–335 

Coagulotoxic cobras: Clinical implications of strong anticoagulant actions of African spitting Naja venoms that are not neutralized by antivenom but are by LY315920 (varespladib) 2018 Mátyás A. Bittenbinder, Christina N. Zdenek, Bianca op den Brouw, Nicholas J. Youngman, James S. Dobson, Arno Naude, Freek J. Vonk, and Bryan G. Fry Toxins 10:516

Failure of Indian antivenom against Bungarus sindanus

Ali Sa, Yang D, Jackson TNW, Undheim EAB, Koludarov I, Wood K, Jones A, Hodgson WC, McCarthy S, Ruder T, Fry BG (2013) Venom proteomic characterization and relative antivenom neutralization of two medically important Pakistani elapid snakes (Bungarus sindanus and Naja naja) Journal of Proteomics.

Reviewer #2: The issue raised by the article by Senji Laxme et al., entitled “Beyond the big four: Venom profiling of the medically important yet neglected Indian snakes reveals disturbing antivenom deficiencies” is crucial. The authors addressed the insufficiency of Indian antivenins manufactured to neutralize the venom from venomous snakes in India. Tested antivenoms did not cover some of the venoms from the 4 main species of Indian venomous snakes ('Big 4') coming from distant regions from the place of manufacture of the antivenoms. Moreover, antivenoms appeared to be ineffective against the venoms from other species of venomous snakes of India ('neglected species').

However, the authors subordinated the antivenom effectiveness to the composition of the venoms used for the manufacture. They deduced that poor efficacy of the antivenoms was based on venom composition variability. If this is probably one of the major reasons for the insufficiency of the antivenoms, there are other substantial causes, e.g. the manufacturing process of the antivenom and adherence to manufacturing standards. In this respect, contrary to what the authors claimed, the manufacture of antivenoms has evolved considerably since their invention in 1894 and we are currently using the fifth generation of antivenoms which are highly purified antibody fragments (at least when the manufacture is appropriate).

Regarding proteomic analysis of venom composition and antivenom neutralization tests, the methodology was reliable and relevant. However, for neutralization tests the use of antivenom dilutions leaded to challenge involving very small amounts of specific antibodies considering a) the low protein concentration of Indian antivenoms, and b) low proportion of specific antibodies relative to the amount of antivenom IgG, the majority of which being not directed against the venom).

The results are clearly presented.

The discussion is not relevant because only limited to composition of venoms as causes of insufficiency of antivenoms.

However, the authors granted too much therapeutic efficacy to the antivenom. It is not a generic drug whose characteristics can be predicted by the choice of the venoms and methods of manufacture, nor even by the results of preclinical tests.

The antivenom is a complex preparation the effectiveness of which stems from many parameters, starting with the venoms used to immunize horses, which the authors seek to evaluate, but also the production of antibodies by the horses, that nobody can anticipate, and the whole manufacturing process that leads to varying results regarding both efficiency and tolerance. The effectiveness of an antivenom is therefore not reproducible from a brand to another. Thus, Authors' decision to test only one antivenom to preserve mice is an ethically virtuous decision but irrelevant on a methodological point of view because it is impossible to deduce the results of neutralization tests of antivenoms from different manufacturers on the basis of the result of a test performed with antivenom from a single manufacturer (Sánchez et al., Toxicon. 2015;106:97-107; Calvete et al. Toxicon. 2016;119:280-8).

Antivenom efficacy varies depending on the manufacturers and even batches from the same manufacturer. Horses' antibody production varies depending both on the immunogenicity of venom components and intrinsic individual immune response. Horses produce on the one hand, different antibodies dependant to the structure of proteins (epitopes), and on the other hand, common antibodies to proteins belonging to the same family even if their mode of action is different (e.g. phospholipases A2 or 3-finger toxins: neurotoxins, cytotoxins, calciseptin, etc.). It has been demonstrated that distinct antigens (or antigens showing different symptoms) can induce a common antibody (Rousselet et al. Eur J Biochem. 1984;140:31-7) and, conversely, single antigen can result in the production of antibodies recognizing many variants of the antigen, including from distinct species (Tremeau et al. FEBS Lett. 1986;208(2):236-40). If the authors are right to write "presence of a wide range of toxin types, capable of exerting variable clinical pathologies following snakebite", which involves a vast range of antibodies, it is also true that many toxins belonging to the same family of proteins (e.g. 3-finger toxins) can be neutralized by a limited number of antibodies. The authors did not seek antigenic kinship between the components of venoms from different species, which would have made it possible to define, perhaps, cross reactions among the antibodies... Amino acids that modify the functioning of the protein (eg substitution of the 49th residue of phospholipase A2), are they included in the epitope that directs the making of the antibody?

In addition, preclinical tests use experimental models (mice) that are not predictable of outcomes in humans. An effective antivenom in mice may not be in humans and vice versa. As a result, the definitive sanction is provided only by clinical trials.

Venom variability has been shown to be almost as important at the specific, geographic, and individual levels (Chippaux et al., Toxicon, 1991: 29: 1279-303), making the manufacture of antivenoms complex and random the effectiveness during envenomation. The response of an antivenom to various venoms from individuals of the same species but from distinct geographical origins, or from distinct individuals from the same region has been studied in detail (Roodt et al. Toxicon. 2011;(7-8):1073-80).

Moreover, the influence of the diet on the composition of the venom remains a hypothesis, and one could just as well say that the composition of the venom is the result of the chance of genetic mutations and gene captures (Casewell et al. Nat Commun. 2012;3:1066), and that it influences the diet, not the other way around. The section on venom variability according to ecology is certainly an interesting hypothesis but not very relevant considering the toxicity of venom since it is not transferable from one animal model to another - and therefore to the human.

The poor neutralization of venoms from neglected species by Indian antivenoms deserves further investigation and, as the authors emphasized, to seek a solution to remedy it. It appeared that the cross-reactions between the antibodies of some of the antivenoms and the venom of species not used for the immunization of horses were restricted. The ELISA tests constitute a first evaluation of the recognition of these venoms by the antibodies. However, only the neutralization tests make it possible to say a) if the antibodies show partial efficacy and what is the titer, b) whether they are toxic antigens or not (at least for the experimental model, i.e. the mouse, and (c) what would be a priori the dosage to consider. It is nevertheless clear that these tests are only indications that make it possible to envisage the inclusion of more or less significant amount of venoms of the neglected species in the cocktail of immunization of horses. The recommendation to Indian manufacturers of antivenoms should be to diversify the sources of their venoms and add some neglected species. A wise choice would be to add not the most common species but those that make it possible to obtain the widest range of antibodies in order to cover -by cross-reactions- the largest number of species.

However, one does not exclude the other, since the final result in humans may not be anticipated.

PLOS authors have the option to publish the peer review history of their article (what does this mean?). If published, this will include your full peer review and any attached files.

Reviewer #1: No

Reviewer #2: Yes: Jean-Philippe Chippaux

---

## [Decision Letter · Decision Letter 1]

1 Nov 2019

Dear Dr Sunagar,

We are pleased to inform you that your manuscript, "Beyond the big four: Venom profiling of the medically important yet neglected Indian snakes reveals disturbing antivenom deficiencies", has been editorially accepted for publication at PLOS Neglected Tropical Diseases.

Before your manuscript can be formally accepted and sent to production you will need to complete our formatting changes, which you will receive in a follow up email. Please note: your manuscript will not be scheduled for publication until you have made the required changes.

IMPORTANT NOTES

* Copyediting and Author Proofs: To ensure prompt publication, your manuscript will NOT be subject to detailed copyediting and you will NOT receive a typeset proof for review. The corresponding author will have one final opportunity to correct any errors when sent the requests mentioned above. Please review this version of your manuscript for any errors.

* If you or your institution will be preparing press materials for this manuscript, please inform our press team in advance at plosntds@plos.org. If you need to know your paper's publication date for media purposes, you must coordinate with our press team, and your manuscript will remain under a strict press embargo until the publication date and time. PLOS NTDs may choose to issue a press release for your article. If there is anything that the journal should know, please get in touch.

*Now that your manuscript has been provisionally accepted, please log into EM and update your profile. Go to http://www.editorialmanager.com/pntd, log in, and click on the "Update My Information" link at the top of the page. Please update your user information to ensure an efficient production and billing process.

*Note to LaTeX users only - Our staff will ask you to upload a TEX file in addition to the PDF before the paper can be sent to typesetting, so please carefully review our Latex Guidelines [http://www.plosntds.org/static/latexGuidelines.action] in the meantime.

Best regards,

Philippe BILLIALD

Associate Editor

José Gutiérrez

Deputy Editor

Reviewer's Responses to Questions

**Key Review Criteria Required for Acceptance?**

**Methods**

-Are the objectives of the study clearly articulated with a clear testable hypothesis stated?

-Is the study design appropriate to address the stated objectives?

-Is the population clearly described and appropriate for the hypothesis being tested?

-Is the sample size sufficient to ensure adequate power to address the hypothesis being tested?

-Were correct statistical analysis used to support conclusions?

-Are there concerns about ethical or regulatory requirements being met?

Reviewer #1: (No Response)

Reviewer #2: (No Response)

**Results**

-Does the analysis presented match the analysis plan?

-Are the results clearly and completely presented?

-Are the figures (Tables, Images) of sufficient quality for clarity?

Reviewer #1: (No Response)

Reviewer #2: (No Response)

**Conclusions**

-Are the conclusions supported by the data presented?

-Are the limitations of analysis clearly described?

-Do the authors discuss how these data can be helpful to advance our understanding of the topic under study?

-Is public health relevance addressed?

Reviewer #1: (No Response)

Reviewer #2: (No Response)

**Editorial and Data Presentation Modifications?**

Reviewer #1: (No Response)

Reviewer #2: (No Response)

**Summary and General Comments**

Reviewer #1: (No Response)

Reviewer #2: (No Response)

PLOS authors have the option to publish the peer review history of their article (what does this mean?). If published, this will include your full peer review and any attached files.

Reviewer #1: No

Reviewer #2: Yes: Jean-Philippe Chippaux

---

## [Editor Report · Acceptance letter]

12 Nov 2019

Dear Dr. Sunagar,

We are delighted to inform you that your manuscript, "Beyond the 'big four': Venom profiling of the medically important yet neglected Indian snakes reveals disturbing antivenom deficiencies," has been formally accepted for publication in PLOS Neglected Tropical Diseases.

Best regards,

Serap Aksoy

Editor-in-Chief

Shaden Kamhawi

Editor-in-Chief
